# Diffusion Blend: Inference-Time Multi-Preference Alignment for Diffusion Models

**Min Cheng**
Texas A&M University

**Fatemeh Doudi**
Texas A&M University

**Dileep Kalathil**
Texas A&M University

**Mohammad Ghavamzadeh**
Qualcomm AI Research

**Panganamala R. Kumar**
Texas A&M University

## Abstract

Reinforcement learning (RL) algorithms have been used recently to align diffusion models with downstream objectives such as aesthetic quality and text-image consistency by fine-tuning them to maximize a single reward function under a fixed KL regularization. However, this approach is inherently restrictive in practice, where alignment must balance multiple, often conflicting objectives. Moreover, user preferences vary across prompts, individuals, and deployment contexts, with varying tolerances for deviation from a pre-trained base model. We address the problem of inference-time multi-preference alignment: given a set of basis reward functions and a reference KL regularization strength, can we design a fine-tuning procedure so that, at inference time, it can generate images aligned with any user-specified linear combination of rewards and regularization, without requiring additional fine-tuning? We propose Diffusion Blend, a novel approach to solve inference-time multi-preference alignment by blending backward diffusion processes associated with fine-tuned models, and we instantiate this approach with three algorithms: DB-MPA for multi-reward alignment, DB-KLA for KL regularization control, and DB-MPA-LS for approximating DB-MPA without additional inference cost. Extensive experiments show that Diffusion Blend algorithms consistently outperform relevant baselines and closely match the performance of individually fine-tuned models, enabling efficient, user-driven alignment at inference-time. The code is available on Github.

## 1 Introduction

Diffusion models, such as Imagen (Saharia et al., 2022a), DALL·E (Ramesh et al., 2022), and Stable Diffusion (Rombach et al., 2022), have demonstrated remarkable capabilities in high-fidelity image synthesis from natural language prompts. However, these models are typically trained on large-scale datasets and are not explicitly optimized for downstream objectives such as semantic alignment, aesthetic quality, or user preference. To address this gap, recent works have proposed reinforcement learning (RL) for aligning diffusion models with task-specific reward functions (Uehara et al., 2024a; Fan et al., 2023; Black et al., 2024), where the core idea is to fine-tune a pre-trained model to maximize a reward, while constraining the update to remain close to the original model via a Kullback–Leibler (KL) regularization. The KL regularization term prevents reward overoptimization (reward hacking), and preserves desirable properties of the pre-trained model (Ouyang et al., 2022; Rafailov et al., 2023) such as sample diversity and visual fidelity (Fan et al., 2023; Uehara et al., 2024b).

While RL fine-tuning has improved alignment in diffusion models, it typically assumes a fixed reward function and regularization weight. This assumption is restrictive in practice, where alignment must balance multiple, often conflicting objectives, such as aesthetics and prompt fidelity, and user preferences vary across prompts, individuals, and deployment contexts. Static fine-tuning with fixed reward combinations cannot accommodate this variability without retraining separate models for each configuration (Wang et al., 2024b; Rame et al., 2023; Lee et al., 2024). Moreover, once trained, the trade-offs are fixed, precluding post-hoc adjustment. Similar issues arise with KL regularization:

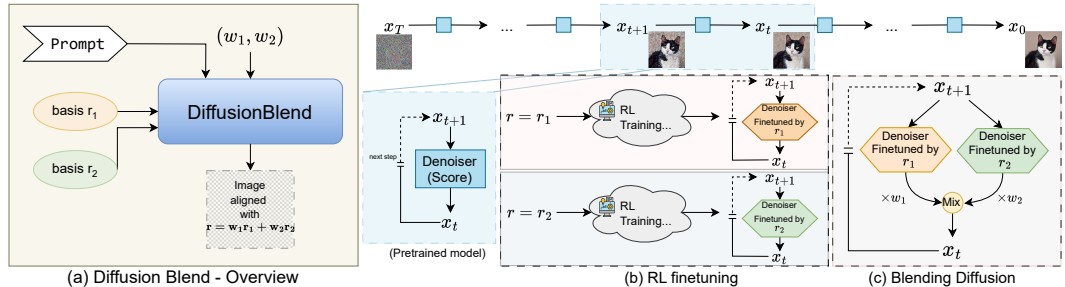

Figure 1: (a). Overview of our Diffusion Blend - Multi Preference Alignment (DB-MPA) Algorithm. Given basis reward functions and any user preference weights $w = (w_1, w_2)$, DB-MPA generates images aligned with combined reward $r(w) = w_1 r_1 + w_2 r_2$. (b) During the fine-tuning stage, DB-MPA gets an RL fine-tuned model corresponding to each reward function. (c) During the inference time, DB-MPA blends (mixes) the backward diffusion corresponding to each fine-tuned model according to the user-specified preference $w$.

insufficient regularization causes reward hacking, while excessive regularization impedes alignment (Uehara et al., 2024b; Liu et al., 2024). In practice, both reward and regularization weights are tuned via grid search, incurring significant computational cost and limiting flexibility.

These limitations motivate the need for a more flexible approach: **inference-time multi-preference alignment**, where the user specifies their preference vector, i.e., weights over a set of basis reward functions such as alignment, aesthetics, or human preference, along with a desired regularization strength that controls deviation from the pre-trained model. Crucially, this alignment must occur without any additional fine-tuning or extensive computation at inference time, which is essential for real-time and resource-constrained settings. Unlike trial-and-error prompt tuning, the ideal solution should offer a principled and computationally efficient solution that can achieve Pareto-optimal trade-offs across multiple preferences. This motivates us to address the following questions:

*Given a set of basis reward functions $(r_i)_{i=1}^m$ and a basis KL regularization weight $\alpha$, can we design a fine-tuning procedure such that when the user specifies their reward or regularization preferences through parameters $w$ and $\lambda$ at inference time, the model generates images aligned with the linear reward combination $r(w) = \sum_{i=1}^m w_i r_i$ and regularization weight $\alpha(\lambda) = \alpha/\lambda$, without requiring additional fine-tuning?* In this work, we answer this question affirmatively and provide constructive solutions to it. Our main contributions are the following:

- We theoretically show that the backward diffusion process corresponding to the diffusion model aligned with reward function $r(w)$ and regularization weight $\alpha$ can be expressed as the backward diffusion process corresponding to the pre-trained model with an additional control term that depends on $r(w)$. We propose an approximation result for this control term, which enables us to express it using the control terms corresponding to fine-tuned diffusion models for the basis reward functions $(r_i)_{i=1}^m$. We also obtain a similar approximation result corresponding to the regularization weight $\alpha$.

- Leveraging the theoretical results we developed, we propose Diffusion Blend - Multi-Preference Alignment **(DB-MPA)** algorithm, a novel approach that will blend the backward diffusion processes corresponding to the basis reward functions appropriately to synthesize a new backward diffusion process during inference-time that will generate images aligned with the reward $r(w)$, where $w$ is specified by the user during the inference-time. Using the same approach, we also propose Diffusion Blend - KL Alignment **(DB-KLA)** algorithm that will generate images aligned with a reward function $r$ and regularization weight $\alpha/\lambda$, where $\lambda$ is specified by the user during the inference time. To reduce the computational overhead associated with DB-MPA, we propose Diffusion Blend - Multi-Preference Alignment- LoRA Sampling **(DB-MPA-LS)** algorithm that addresses the increased inference time issue while maintaining similar performance.

- We provide extensive experimental evaluations using the Stable Diffusion (Rombach et al., 2022) baseline model, multiple basis reward functions, regularization weights, standard prompt sets, and demonstrate that our diffusion blend algorithms outperform multiple relevant baseline algorithms, and often achieve a performance close to the empirical upper bound obtained by an individually fine-tuned model for specific $w$ and $\lambda$.

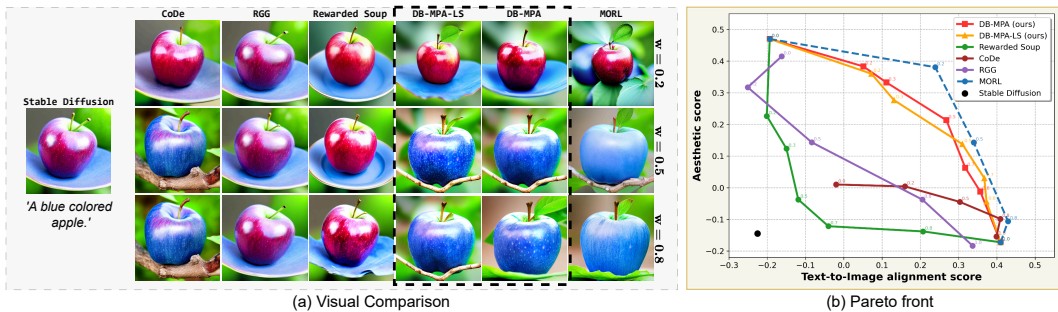

(a) Visual Comparison   (b) Pareto front

Figure 2: Comparison of DB-MPA with baselines: Stable Diffusion v1.5 (Rombach et al., 2022), CoDe (Singh et al., 2025), RGG (Chung et al., 2023), rewarded soup (RS) (Rame et al., 2023), and MORL (Roijers et al., 2013). Note that MORL is included only to illustrate the maximum achievable performance by an oracle algorithm. See section 2 for details. For arbitrary preference weight $w$, algorithms generate images aligned with $r(w) = wr_1 + (1 - w)r_2$, where $r_1$ is text-to-image alignment and $r_2$ is aesthetics. (a) Images for $w \in \{0.2, 0.5, 0.8\}$. (b) Pareto-front comparison. DB-MPA significantly outperforms baselines and approaches the MORL upper bound.

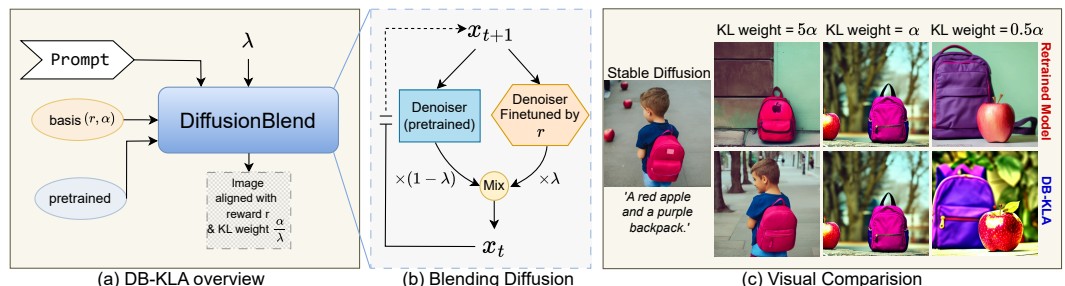

(a) DB-KLA overview   (b) Blending Diffusion   (c) Visual Comparision

Figure 3: (a) Overview of our Diffusion Blend-KL Alignment (DB-KLA) Algorithm. Given an RL fine-tuned model for reward $r$ with KL weight $\alpha$, DB-KLA generates images aligned with KL weight $\alpha/\lambda$ for any user-specified modification factor $\lambda$. (b) During inference, DB-KLA blends the backward diffusion of the fine-tuned and pretrained models according to $\lambda$, which can be larger than 1. (c) Visual comparisons with $\lambda$-specific RL retrained models using text-to-image-alignment reward and $\lambda \in \{0.2, 1.0, 2.0\}$. DB-KLA achieves smooth control by adjusting the effective distance from the pre-trained model via $\lambda$, generating images similar to $\lambda$-specific RL models without additional fine-tuning.

## 2 RELATED WORK

*Finetuning-based algorithms:* Prior works align diffusion models via reward-guided finetuning. Rewards-in-context (Yang et al., 2024) conditions on multiple reward types, DRaFT (Clark et al., 2024) uses weighted combinations during training, and (Hao et al., 2023) applies RL with alignment–aesthetic trade-offs. Parrot (Lee et al., 2024) leverages prompt expansion, while Calibrated DPO (Lee et al., 2025) aggregates multiple reward models. Rewarded Soup (RS) (Rame et al., 2023) is closest to us, linearly combining parameters from reward-specific models, whereas our DB-MPA blends backward diffusion trajectories in a principled way.

*Guidance algorithms:* Gradient-based methods (Chung et al., 2023; Yu et al., 2023; Song et al., 2023; Bansal et al., 2023; He et al., 2024; Ye et al., 2024) add reward gradients at each reverse diffusion step, and can handle multiple objectives (Han et al., 2023; Kim et al., 2025; Ye et al., 2024). They require differentiable rewards and Tweedie-based approximations (Efron, 2011), leading to noise and high cost. Gradient-free methods instead generate multiple candidates and select high-reward samples (Mudgal et al., 2024; Gui et al., 2024; Beirami et al., 2024), or use particle/value-guided search (Li et al., 2024; Singhal et al., 2025; Singh et al., 2025). These avoid gradients but demand heavy sampling and reward access.

*Multi-Objective RL (MORL):* Approaches (Roijers et al., 2013; Yang et al., 2019; Zhou et al., 2022; Rame et al., 2023) fine-tune a separate model for each preference or regularization weight. While theoretically optimal, inference-time RL is infeasible; even covering the weight space requires exponentially many models. We thus treat MORL only as an oracle baseline.

*Multi-preference alignment in LLMs:* Works such as (Rame et al., 2023; Jang et al., 2023; Shi et al., 2024; Wang et al., 2024b; Guo et al., 2024; Zhong et al., 2024b; Wang et al., 2024a) extend RL finetuning to LLMs. For KL-regularized alignment, DeRa (Liu et al., 2024) controls alignment by combining logits from aligned and reference models. Our diffusion blend methods are inspired by these but introduce inference-time preference alignment specifically for diffusion models.

*LoRA composition for image generation:* Recent work on multi-concept fusion in diffusion models (Zhong et al., 2024a; Zou et al., 2025) focuses on composing pretrained LoRA modules using fixed, heuristic scheduling to mitigate degradation when multiple concepts are combined. In contrast, our DB-MPA framework blends reward-aligned diffusion trajectories, enabling principled and interpretable trade-offs rather than heuristic mixing.

## 3 PRELIMINARIES AND PROBLEM FORMULATION

**Diffusion model and pre-training:** A diffusion model (Ho et al., 2020; Song et al., 2021) approximates an unknown data distribution $p_{\text{data}}$ by an iterative approach. It consists of a forward process and a backward process. In the forward process, a clean sample from the data distribution $p_{\text{data}}$ is progressively corrupted by adding Gaussian noise at each timestep, ultimately transforming the data distribution into pure noise. The reverse process involves training a denoising neural network to iteratively remove the added noise and reconstruct samples from the original data distribution. The forward process is typically represented by the stochastic differential equation (SDE), $\mathrm{d}x_t = -\frac{1}{2}\beta(t)x_t\mathrm{d}t + \sqrt{\beta(t)}\mathrm{d}w_t, \ \forall t \in [0, T]$,

where $x_0 \sim p_{\text{data}}$, $\beta(t)$ is a predefined noise scheduling function, and $w_t$ represents a standard Wiener process. The reverse process of this SDE is given by (Anderson, 1982; Song et al., 2021)

$$\mathrm{d}x_t = [-\frac{1}{2}\beta(t)x_t - \beta(t)\nabla_{x_t}\log p_t(x_t)]\mathrm{d}t + \sqrt{\beta(t)}\mathrm{d}w_t, \ \forall t \in [T, 0], \tag{1}$$

where $p_t$ denotes the marginal probability distribution of $x_t$, $x_T$ is sampled according to a standard Gaussian distribution, and $\nabla_{x_t}\log p_t(x_t)$ represents the *score function* that guides the reverse process. Since the marginal density $p_t$ is unknown, the score function is estimated by a neural network $s_\theta$ through minimizing score-matching objective (Song et al., 2021) given by the optimization problem, $\arg\min_\theta \ \mathbb{E}_{t \sim U[0,T]}\mathbb{E}_{x_0 \sim p_{\text{data}}}\mathbb{E}_{x_t \sim p_t(\cdot|x_0)}\left[\lambda(t)\left\|\nabla_{x_t}\log p_t(x_t|x_0) - s_\theta(x_t, t)\right\|^2\right]$, where $s_\theta(x_t, t)$ is a neural network parameterized by $\theta$ that approximates the score function and $\lambda(t)$ is a weighting function. In the following, we denote the pre-trained diffusion model by the (backward) SDE

$$\mathrm{d}x_t = f^{\text{pre}}(x_t, t)\mathrm{d}t + \sigma(t)\mathrm{d}w_t, \ \forall t \in [T, 0], \tag{2}$$

where $f^{\text{pre}}(x_t, t)$ denotes the term $[-\frac{1}{2}\beta(t)x_t - \beta(t)\nabla_{x_t}\log p_t(x_t)]$ in eq. (1), and $\sigma(t)$ denotes the noise scheduling function. We will use $p_t^{\text{pre}}$ to denote the distribution of $x_t$ according to the pre-trained diffusion model given by the backward SDE in eq. (2).

**RL fine-tuning of pre-trained diffusion model:** Diffusion models are pre-trained to learn the score function, and are not trained to additionally maximize downstream rewards such as aesthetic score. Aligning a pre-trained diffusion model with a given reward function $r(\cdot)$ can be formulated as the optimization problem $\max_{p_0} \mathbb{E}_{x_0 \sim p_0}[r(x_0)]$, with the initialization $p_0^{\text{pre}}$.

However, this can lead to reward over-optimization, disregarding the qualities of data-generating distribution $p_0^{\text{pre}}$ learned during the pre-training (Fan et al., 2023). To avoid this issue, similar to the LLM fine-tuning (Ouyang et al., 2022), a KL-divergence term between the pre-trained and fine-tuned models is included as a regularizer to the RL objective (Fan et al., 2023), resulting in the *diffusion model alignment problem*:

$$\max_{p_0} \ \mathbb{E}_{x_0 \sim p_0}[r(x_0)] - \alpha\mathrm{KL}(p_0\|p_0^{\text{pre}}), \tag{3}$$

where $\alpha$ is the KL regularization weight. Since directly evaluating KL divergence between $p_0$ and $p_0^{\text{pre}}$ is challenging, it is to upperbounded it by the sum of the KL divergences between the conditional distributions at each step (Fan et al., 2023), resulting in the fine-tuning objective

$$\max_{(p_t)_{t=T}^0} \mathbb{E}[r(x_0) - \alpha \sum_{t=T}^1 \mathrm{KL}(p_t(\cdot|x_t)\|p_t^{\mathrm{pre}}(\cdot|x_t))], \tag{4}$$

where the expectation is taken w.r.t. $\Pi_{t=T}^1 p_t(x_{t-1}|x_t)$, and $x_T \sim p_T$. While the optimization is over a sequence of distributions, $(p_t)_{t=T}^0$, in practice, we learn only the score function parameter $\theta$ which will induce the distributions $p_t$ as $p_t^\theta$. We assume that this fine-tuning objective will solve eq. (3) approximately, which will result in a fine-tuned model aligned with $r$ and $\alpha$. Similar to the notation in eq. (2), we represent the backward diffusion process corresponding to this fine-tuned model as

$$\mathrm{d}x_t = f^{(r,\alpha)}(x_t, t)\mathrm{d}t + \sigma(t)\mathrm{d}w_t, \ \forall t \in [T, 0], \tag{5}$$

where $f^{(r,\alpha)}$ depends on $r, \alpha$. The exact form of $f^{(r,\alpha)}$ is derived in proposition 1.

The fine-tuning problem in eq. (4) is typically solved using RL by formulating it as an entropy-regularized Markov Decision Process (MDP) (Fan et al., 2023; Uehara et al., 2024a). The state $\mathcal{S}$ and action $\mathcal{A}$ spaces are defined as the set of all images $\mathcal{X}$. The transition dynamics at each step $t$ is deterministic, $P_t(s_{t+1} \mid s_t, a_t) = \delta(s_{t+1} = a_t)$. The reward function is non-zero only for $t = T$ and $r(s_t) = 0, \forall t \in \{0, \ldots, T-1\}$. The policy $\pi_t : \mathcal{S} \to \Delta(\mathcal{A})$ is modeled as a Gaussian distribution, matching with the discretization of the reverse process given in eq. (2). To align with the diffusion model notation, we define $s_t = x_{T-t}, a_t = x_{T-t-1}$, and $\pi_t(a_t|s_t) = p_{T-t-1}(x_{T-t-1}|x_{T-t})$. The initial distribution of the state $s_0 = x_T$ is standard Gaussian.

**Problem: Inference-Time Multi-Prefence Alignment:** The main issue with the standard RL fine-tuning (eq. (3)-eq. (4)) is that the fine-tuned model is optimized for a fixed $(r, \alpha)$, and the model is unchangeable after fine-tuning. So, it is not possible to generate optimally aligned data samples for another reward $r$ or regularization weight $\alpha$ at inference time. To address this problem, we follow the standard multi-objective RL formalism (Yang et al., 2019; Zhou et al., 2022), where we consider a multi-dimensional reward $(r_1, r_2, \ldots, r_m)$ and assume that the reward is represented by a linear scalarization $r(w) = \sum_{i=1}^m w_i r_i, w \in \Delta_m$, where $\Delta_m$ is the $m$-dimensional simplex. At inference time, the user communicates their preferences by specifying the reward function weight $w$. We also assume that the user specifies a regularization modification factor $\lambda$ to propose an effective regularization weight $\alpha(\lambda) = \alpha/\lambda$. In this paper, we address the following problem:

*How do we solve the alignment problem in eq.* (3) *for arbitrary reward function $r(w)$ and regularization weight $\alpha(\lambda)$ without additional fine-tuning at inference time? Note that $w$ and $\lambda$ are user-specified values at inference time.*

## 4 DIFFUSION BLEND ALGORITHM

The alignment problem in eq. (3) is the same as the one used for LLMs (Rafailov et al., 2023; Liu et al., 2024). Recent results in LLMs, especially those that use the direct preference optimization approach (Rafailov et al., 2023), have leveraged the following closed-form solution to eq. (3) to develop alignment algorithms:

$$p^{\mathrm{tar}}(x_0) = p^{\mathrm{pre}}(x_0) \cdot \exp\big(r(x_0)/\alpha\big) \ / \ Z \ , \tag{6}$$

where $Z$ is a normalization constant. However, in the case of diffusion models, directly sampling from $p^{\mathrm{tar}}$ is infeasible for two reasons. First, computing $Z$ is intractable, as it requires evaluating an integral over a high-dimensional continuous space. Second, unlike in the case of LLM where $p^{\mathrm{pre}}(x)$ for any token $x$ is available at the output layer of the LLM, diffusion models do not offer an explicit way to evaluate $p^{\mathrm{pre}}(x)$ for an arbitrary $x$. Instead, we can only sample from $p^{\mathrm{pre}}$ by running a backward SDE. In other words, we will not be able to directly sample from $p^{\mathrm{tar}}$ by tilting $p^{\mathrm{pre}}$ as given in eq. (6), even if the value of $Z$ is known. RL fine-tuning achieves sampling from $p^{\mathrm{tar}}$ by learning a model that can synthesize the backward diffusion specified by $f^{(r,\alpha)}$. However, this would suggest an extensive RL fine-tuning for different values of $w$ and $\lambda$ in $r(w)$ and $\alpha(\lambda)$. To address this, we describe an interesting mapping between $f^{\mathrm{pre}}$ and $f^{(r,\alpha)}$, which we will then exploit to solve the inference-time multi-preference alignment problem without naive extensive fine-tuning.

Let $x_0^{\mathrm{pre}} \sim p^{\mathrm{pre}}(\cdot)$, $x_0^{\mathrm{tar}} \sim p^{\mathrm{tar}}(\cdot)$, and $(\epsilon_t)_{t=1}^T$ be an independent, zero mean Gaussian noise sequence with probability distribution $p_{\epsilon_t}$, i.e., $\epsilon_t \sim p_{\epsilon_t}(\cdot)$. Consider the forward noise processes

$x_t^{\mathrm{pre}} = x_0^{\mathrm{pre}} + \epsilon_t$ and $x_t^{\mathrm{tar}} = x_0^{\mathrm{tar}} + \epsilon_t$, and let $p_t^{\mathrm{pre}}$ and $p_t^{\mathrm{tar}}$ be the marginal distributions of $x_t^{\mathrm{pre}}$ and $x_t^{\mathrm{tar}}$, respectively. As standard in the diffusion model literature (Song et al., 2021; Ho et al., 2020), we assume that under the forward noise process, the distributions of $x_T^{\mathrm{pre}}$ and $x_T^{\mathrm{tar}}$ are Gaussian. Let $p_{0|t}^{\mathrm{pre}}$ denote the conditional distribution of $x_0^{\mathrm{pre}}$ given $x_t^{\mathrm{pre}}$. Then, we have the following result.

**Proposition 1.** *Let $f^{(r,\alpha)}$ and $f^{\mathrm{pre}}$ be as specified in eq. (5) and eq. (2), respectively. Then, $f^{(r,\alpha)}(x_t, t) = f^{\mathrm{pre}}(x_t, t) - \beta(t) u^{(r,\alpha)}(x_t, t)$, where*

$$u^{(r,\alpha)}(x_t, t) = \nabla_{x_t} \log \, \mathbb{E}_{x_0 \sim p_{0|t}^{\mathrm{pre}}(\cdot|x_t)} \left[ \exp\left( \frac{r(x_0)}{\alpha} \right) \right]. \tag{7}$$

*Remark* 1. We prove proposition 1 following the SDE interpretation of diffusion models (Song et al., 2021), and by showing that two SDEs initialized at time $t = 0$ at $p^{\mathrm{pre}}$ and $p^{\mathrm{tar}}$, and sharing the same forward noise injection process, can be reversed similarly. In particular, we show that the key parameters of the corresponding two reverse SDEs remain unchanged, except that the latter includes an additional control term $u^{(r,\alpha)}$ in the score function. We note that (Uehara et al., 2024b) derives a similar result by analyzing the RL objective in eq. (4) and leveraging results from stochastic optimal control. In contrast, our approach analyzes the original alignment objective in eq. (3) and derives a simpler, first-principles proof without relying on stochastic optimal control theory, under the standard mild assumption that for large $T$, the terminal distribution of both forward SDEs is Gaussian.

We now consider an approximation $\bar{u}^{(r,\alpha)}$ to $u^{(r,\alpha)}$, motivated by the Jensen gap approximation idea that has been successfully utilized in algorithms for noisy image inverse problems (Chung et al., 2023; Rout et al., 2023; 2024). Let $x$ be a random variable with distribution $p$. For a non-linear function $f$, the Jensen gap is defined as $\mathbb{E}[f(x)] - f(\mathbb{E}[x])$. In our case, we interchange the expectation $\mathbb{E}[\cdot]$ and the nonlinear function $\exp(\cdot)$ in $u^{(r,\alpha)}$ to obtain the following approximation:

$$u^{(r,\alpha)}(x, t) = \bar{u}^{(r,\alpha)}(x, t) + \Delta^{(r,\alpha)}(x, t), \quad \text{where} \quad \bar{u}^{(r,\alpha)}(x, t) = \nabla_x \mathbb{E}_{x_0 \sim p_{0|t}^{\mathrm{pre}}(\cdot|x)} \left[ \frac{r(x_0)}{\alpha} \right]. \tag{8}$$

We now prove an upper-bound on the approximation error $\Delta^{(r,\alpha)}(x, t)$ in eq. (8) under certain reasonable assumptions. Refer to appendix A.2 for more details.

**Lemma 1.** *Assume $\frac{r(x_0^{\mathrm{pre}})}{\alpha} = \tilde{r}(x_t^{\mathrm{pre}}, t) + \eta(\omega, x_t^{\mathrm{pre}}, t)$, where $\eta(\omega, x_t^{\mathrm{pre}}, t)$ specifies the randomness induced by the backward diffusion process. Denote $p_{R|t}$ as the conditional distribution of the random variable $R := r(x_0^{\mathrm{pre}})/\alpha$ given $x_t^{\mathrm{pre}}$. Then,*

$$|\Delta^{(r,\alpha)}(x, t)| \leq L_{t,1}(x) \times L_{t,2}(x) + L_{t,3}(x), \qquad where$$

$$L_{t,1}(x) = \sqrt{\mathbb{E}[\|\nabla_x \eta(\omega, x_t^{\mathrm{pre}}, t)\|_2^2 \mid x_t^{\mathrm{pre}} = x]}, \quad L_{t,2}(x) = \frac{\sqrt{\mathrm{Var}(\exp(r(x_0^{\mathrm{pre}})/\alpha) \mid x_t^{\mathrm{pre}} = x)}}{\mathbb{E}[\exp(r(x_0^{\mathrm{pre}})/\alpha) \mid x_t^{\mathrm{pre}} = x]},$$

$$L_{t,3}(x) = (1 + \frac{1}{\alpha}) \cdot \sup_r \|\nabla_x \log p_{R|t}(r \mid x_t^{\mathrm{pre}} = x) + \nabla_r \log p_{R|t}(r \mid x_t^{\mathrm{pre}} = x)\|.$$

*Remark* 2. In lemma 1, the term $L_{t,1}$ quantifies the local Lipschitz sensitivity of the stochastic component of $R$ with respect to changes in $x_t$; $L_{t,2}$ denotes the conditional coefficient of variation (i.e., the ratio of standard deviation to mean) of $R$ given $x_t$; and $L_{t,3}$ measures the deviation of the conditional distribution $p(x_0 \mid x_t)$ from a pure shift family. A shift family (or location family) refers to a class of conditional distributions where changing the conditioning variable results in a simple translation of the distribution without altering its shape, e.g., $P(X = x|Y = y) = P(X = x - y)$ (Casella and Berger, 2024). For a perfect shift family, we may write $L_{t,3} \equiv 0$. In diffusion models, as $t$ gets closer to 0, $p_{0|t}^{\mathrm{pre}}(x_0|x_t)$ becomes more deterministic and concentrates around $x_t$, with variation of $x_t$ reducing to a mean shift, resulting $L_{t,3}$ to get closer to 0. When the reward function is more predictable from the noisy image $x_t$ or $t$ becomes closer to 0, both $L_{t,1}$ and $L_{t,2}$ will be small. Note that $L_{t,2}$ and $L_{t,3}$ will increase when the regularization coefficient $\alpha$ becomes very small, suggesting that our algorithm might fail when we decrease $\alpha$ dramatically. We note that (Uehara et al., 2024b) exchanges the order of $\mathbb{E}[\cdot]$ and $\exp(\cdot)$ under the assumption $r(x_0) = k(x_t) + \epsilon$, where $\epsilon$ is an independent noise term. This is consistent with our result, as it would be easy to derive $L_{t,1} \equiv L_{t,3} \equiv 0$ under their assumption.

The key motivation behind the approximation in eq. (8) is that we can now leverage the linearity of expectation available in $\bar{u}^{(r,\alpha)}$ to approximate $f^{(r(w),\alpha(\lambda))}$ in terms of $f^{(r_i,\alpha)}$, $i = 1, \ldots, m$.

**Lemma 2.** *Let $f^{(r,\alpha)}$ be as specified in eq.* (5). *Then, we have*

$$f^{(r(w),\alpha)}(x_t, t) = \sum_{i=1}^{m} w_i f^{(r_i,\alpha)}(x_t, t) + \beta(t)\left(\sum_{i=1}^{m} w_i \Delta^{(r_i,\alpha)}(x, t) - \Delta^{(r(w),\alpha)}(x, t)\right), \qquad (9)$$

$$f^{(r,\alpha(\lambda))}(x_t, t) = (1 - \lambda)f^{\mathrm{pre}}(x_t, t) + \lambda f^{(r,\alpha)}(x_t, t) + \beta(t)\left(\lambda\Delta^{(r,\alpha)}(x, t) - \Delta^{(r,\alpha(\lambda))}(x, t)\right). \tag{10}$$

Using the result in lemma 2, we now introduce our diffusion blend algorithms, with pseudo code provided in appendix B.2.

**Diffusion Blend-Multi-Preference Alignment (DB-MPA) Algorithm:** Our goal is to solve the alignment problem in eq. (3) for an arbitrary reward function $r(w)$ with user-specified parameter $w$, without additional fine-tuning at inference time. This is equivalent to obtaining the diffusion term $f^{(r(w),\alpha)}$ and running the backward SDE in eq. (5). At the fine-tuning stage (before deployment), we independently fine-tune the pre-trained model for each reward $(r_i)_{i=1}^{m}$ with fixed $\alpha$, obtaining $m$ RL fine-tuned models $(\theta_i^{\mathrm{rl}})_{i=1}^{m}$ by solving the fine-tuning objective in eq. (3). At inference, we use lemma 2 to approximate $f^{(r(w),\alpha)}(x_t, t) \approx \sum_{i=1}^{m} w_i f^{(r_i,\alpha)}(x_t, t)$, where each $f^{(r_i,\alpha)}$ is computed using the RL fine-tuned model $\theta_i^{\mathrm{rl}}$. We then generate samples by running the backward SDE in eq. (5).

**Diffusion Blend-KL Alignment (DB-KLA) Algorithm:** Our goal is to solve the alignment problem in eq. (3) for an arbitrary regularization weight $\alpha(\lambda)$ with user-specified parameter $\lambda$, without additional fine-tuning at inference time. This is equivalent to running the backward diffusion in eq. (5) with $f^{(r,\alpha(\lambda))}$. At fine-tuning, we fine-tune the pre-trained model for reward $r$ and regularization weight $\alpha$, obtaining RL fine-tuned model $\theta^{\mathrm{rl}}$ from $\theta^{\mathrm{pre}}$. At inference, we use lemma 2 to approximate $f^{(r,\alpha(\lambda))}(x_t, t) \approx (1 - \lambda)f^{\mathrm{pre}}(x_t, t) + \lambda f^{(r,\alpha)}(x_t, t)$, where $f^{\mathrm{pre}}$ and $f^{(r,\alpha)}$ are computed using $\theta^{\mathrm{pre}}$ and $\theta^{\mathrm{rl}}$, respectively. We then generate samples by running the backward SDE in eq. (5).

### 4.1 INFERRENCE TIME EFFICIENT VARIANT

While DB-MPA and DB-KL successfully achieve reward alignment through score merging, this approach requires evaluating all $m$ diffusion models at each denoising step, resulting in $m\times$ computational overhead during inference. To address this limitation, we propose DB-MPA-with-LoRA-Sampling (DB-MPA-LS), a novel algorithm that approximates the score merging process by randomly sampling reward fine-tuned LoRA adapters at each denoising step with probabilities proportional to their assigned weights. This approach reduces the inference cost to that of the original pre-trained Stable Diffusion model, eliminating the multiplicative overhead inherent in inference-time realignment methods, including our DB-MPA and existing LLM variants Liu et al. (2024); Shi et al. (2024). Note that this sampling approximation cannot be applied to DB-KL since the KL reweighting terms may be negative. The key insight is that unlike LLM realignment which mixes probabilities over discrete and finite tokens, diffusion models operate through continuous stochastic processes where the noise-adding nature enables a different mathematical treatment, which we show by the following proposition.

**Proposition 2.** *For the Lipschitz continuous functions $f_1$ and $f_2$, the following two SDE have the same marginal probability $p_{X_t^1} = p_{X_t^2}$ for $\forall\, t \in [0, T]$. SDE 1 is $dX_t^1 = (af_1(X_t^1) + (1 - a)f_2(X_t^1))dt + \sigma(t)d\omega_t$, with $X_0 \sim p_0$, $t \in [0, T]$, and $\{\omega_t\}$ being the Winner process. SDE 2 is $dX_t^2 = (Y_t f_1(X_t^2) + (1 - Y_t)f_2(X_t^2))dt + \sigma(t)d\omega_t$, with $X_0 \sim p_0$, $t \in [0, T]$, and $\{\omega_t\}$ being the Winner process, where $Y_t$ is a Bernoulli random variable with probability $a$ to be 1 and probability $1 - a$ to be 0, and $Y_t$ is independent of $\{X_t^2\}_t$ and $Y_s$ for any $s \neq t$.*

*Remark* 3. Without loss of generality, we present the theoretical result for the two-reward case ($m = 2$), as the extension to arbitrary finite $m$ rewards follows straightforwardly by replacing the Bernoulli variable with a categorical random variable. Details of the proof are in appendix A.4.

## 5 EXPERIMENTS

In this section, we present comprehensive experimental evaluations that demonstrate the superior performance of our DB-MPA and DB-KLA algorithms compared to the baseline models.

**Reward models.** We use four reward models in our experiments: $(i)$ *ImageReward* (Xu et al., 2024), which measures the text-image alignment and is used in its original form; $(ii)$ *VILA* (Ke et al., 2023), which measures the aesthetic quality of generated images and outputs in $[0, 1]$, is rescaled to $[-2, 2]$ via $r \mapsto 4r - 2$ to normalize its influence relative to other rewards; and $(iii)$ *PickScore* (Kirstain et al., 2023), which measures how well an image generated from a text prompt aligns with human preferences, is shifted by $-19$ to match the scale of other rewards. $(iv)$ We further test our algorithm on a *JPEG compressibility* reward, which opposes aesthetics by favoring smooth images, enabling analysis of adversarial alignment.

**Baselines:** We compare the performance of our algorithms with the following baseline algorithms: $(i)$ Rewarded Soup (RS) (Rame et al., 2023), $(ii)$ CoDe (Singh et al., 2025), a gradient-free guidance algorithm with look-ahead search, where we use $N = 20$ particles for the search and $B = 5$ look-ahead steps. $(iii)$ Reward gradient-based guidance (RGG) (Chung et al., 2023; Kim et al., 2025), and $(iv)$ Multi-Objective RL (MORL) (Rame et al., 2023; Wu et al., 2023). Details of these baselines are given in appendix B.1. Note that we report MORL performance only as an oracle baseline, illustrating the maximum alignment achievable.

**Prompt datasets:** We use two benchmark datasets in our experiments. $(i)$ We first select the color subset from DrawBench (Saharia et al., 2022b), comprising 25 prompts out of the full 183 prompts across 11 categories. RL training can reliably converge at this small-scale setup, which aligns with our theoretical assumption that RL converges to the closed-form solution of eq. (3), while our DB algorithm is interpolating between those optimal solutions under individual reward functions. For evaluation, we generate a test set of 1,000 prompts using a pipeline similar to GenEval with random color-object combinations. The candidate lists of colors and objects are generated by GPT-4 (Achiam et al., 2023) to be semantically similar to the training set (implementation details and full list in our code). $(ii)$ We also validate performance of DB on the GenEval dataset (Kirstain et al., 2023), which contains 550 prompts across six compositional tasks: single objects, two objects, counting, colors, spatial positions, and color attribution. An additional 700 test prompts are generated using the official GenEval prompt generation script.

**Training and evaluation details:** We use Stable Diffusion v1.5 (SDv1.5) (Rombach et al., 2022) as the base model for our experiments, which is a text-to-image model capable of generating high-resolution images. We use the DPOK algorithm (Fan et al., 2023) for RL fine-tuning. For the experimental results given in the main paper, we use the 1000 test prompts. Details of the implementation, including training configurations and hyperparameters, are given in appendix B.3.

### 5.1 DB-MPA ALGORITM RESULTS

Table 1: Quantitative comparison of DB-MPA and baseline methods

| | | SD | | MORL | | DB-MPA | | DB-MPA-LS | | RS | | CoDe | | RGG | |
|---|---|---|---|---|---|---|---|---|---|---|---|---|---|---|---|
| | | $r_1$ | $r_2$ | $r_1$ | $r_2$ | $r_1$ | $r_2$ | $r_1$ | $r_2$ | $r_1$ | $r_2$ | $r_1$ | $r_2$ | $r_1$ | $r_2$ |
| | $w{=}0.0$ | -0.22 | -0.14 | -0.19 | **0.47** | -0.19 | **0.47** | -0.19 | **0.47** | -0.19 | **0.47** | **-0.02** | 0.01 | -0.16 | 0.42 |
| | $w{=}0.2$ | -0.22 | -0.14 | **0.24** | **0.38** | 0.05 | **0.38** | 0.07 | 0.36 | -0.20 | 0.23 | 0.16 | 0.00 | -0.25 | 0.32 |
| Reward (↑) | $w{=}0.5$ | -0.22 | -0.14 | **0.34** | 0.14 | 0.27 | **0.21** | 0.31 | 0.14 | -0.12 | -0.04 | 0.30 | -0.04 | -0.08 | 0.14 |
| | $w{=}0.8$ | -0.22 | -0.14 | **0.43** | -0.11 | 0.36 | **-0.01** | 0.37 | -0.04 | 0.21 | -0.14 | 0.41 | -0.10 | 0.21 | -0.04 |
| | $w{=}1.0$ | -0.22 | -0.14 | **0.41** | -0.17 | **0.41** | -0.17 | **0.41** | -0.17 | **0.41** | -0.17 | 0.40 | **-0.15** | 0.34 | -0.18 |
| Inference Time (↓ sec/img) | | **5.46** | | **5.46** | | 11.11 | | 5.64 | | **5.46** | | 185.26 | | 121.58 | |

We first consider two reward functions ($m = 2$), with $r_1$ as ImageReward (Xu et al., 2024), which measures text-image alignment, and $r_2$ as VILA (Kirstain et al., 2023), which measures aesthetics. During the inference time, the user specifies a preference weight $w \in [0, 1]$ to obtain data samples aligned to the reward $wr_1 + (1 - w)r_2$. We fix $\alpha = 0.1$ for these experiments, where $\alpha$ is the KL weight.

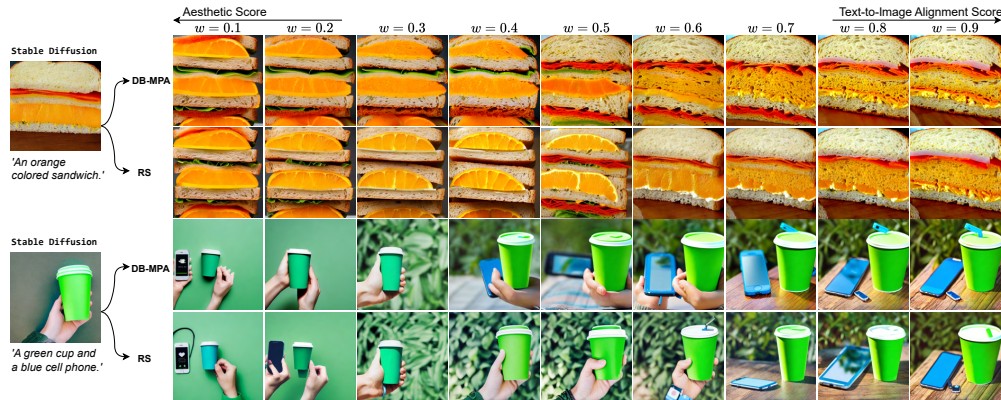

Figure 4: Illustration of the smooth control of DB-MPA to generate images aligned with $r(w)$ for any $w \in [0, 1]$. DB-MPA generates images that are better aligned with both rewards, especially for $w \in [0.4, 0.8]$. RS generates images with wrong interpretation objects (orange) or missing objects (cellphone).

In fig. 2(b), we present the Pareto-front of DB-MPA against baseline algorithms under the *Short-DrawBench* setting, evaluated on the 1k test prompts. For DB-MPA and RS, we evaluate the performance for $w \in \{0.1, \ldots, 0.9\}$. For other baselines, we evaluate their performance for $w \in \{0.2, 0.5, 0.8\}$ due to their high inference cost. Our DB-MPA algorithm consistently outperforms the baseline in all these experiments and achieves a Pareto-front very close to that of MORL, which represents the theoretical optimum obtainable by RL fine-tuning. We further evaluate the lightweight variant DB-MPA-LS, which achieves nearly identical Pareto performance while matching the inference speed of standard Stable Diffusion. As shown in fig. 11, the outputs of DB-MPA and DB-MPA-LS are also visually close. In table 1, we provide a quantitative comparison of this Pareto-front result. If we take the weighted reward $r(w)$ as a metric of comparison, for $w = 0.5$, DB-MPA (0.42) has close performance to DB-MPA-LS 0.39 and outperforms RS, CoDe, and RGG by **3.92×**, **1.95×**, and **1.33×**, respectively. table 1 also shows the inference time comparison. DB-MPA uses two fine-tuned models, making its inference time about twice that of Stable Diffusion. CoDe and RGG, though single-model methods, incur far higher costs due to multi-particle sampling and gradient steps. The lightweight DB-MPA-LS matches DB-MPA's performance while running at nearly the same speed as Stable Diffusion. As analyzed in appendix C.4, DB-MPA-LS maintains performance close to DB-MPA as the number of rewards increases, with DB/LS ratios of the average gains under each fixed reward setting being 1.02, 1.15, 1.08 for 2, 3, 4 rewards, while RS performance drops significantly (DB/RS = 3.3, 6.4, 8.2).

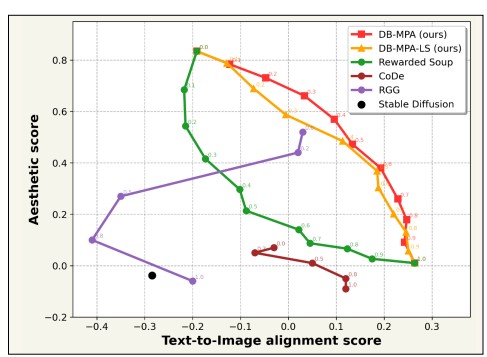

Figure 5: Pareto-front comparison of DB-MPA and DB-MPA-LS algorithm with other baselines, evaluated on GenEval test prompt set.

We further scale DB-MPA to the GenEval benchmark by fine-tuning models on all 550 prompts. Evaluation on the 700 held-out GenEval test prompts (table 4 and its corresponding Pareto boundary in fig. 5) shows that both DB-MPA and DB-MPA-LS consistently dominate the baselines across all preference weights. We also verify scalability on a larger base model (SDXL) in appendix C.6, where a single-prompt experiment demonstrates consistent performance gains.

fig. 4 illustrates the smooth control of DB-MPA in generating images aligned with $r(w)$ for any $w \in [0, 1]$. We use RS as the primary baseline due to the high inference cost of other methods. DB-MPA consistently produces images better aligned with both $r_1$ and $r_2$, particularly for $w \in [0.4, 0.8]$. Additional results in appendix C.8 and appendix C.9 further confirm that DB-MPA generates images of higher quality than the baseline and comparable to those from MORL.

We further examine a challenging case of conflicting rewards by introducing the *JPEG compressibility* reward, which favors smooth, low-detail images and directly opposes the VILA aesthetic reward that emphasizes fine-grained visual quality. As shown in appendix C.3, DB-MPA maintains superior performance even under this adversarial setting. We also extend our study to the three-reward case ($m = 3$) by incorporating *PickScore* (Kirstain et al., 2023), which measures human preference alignment. The corresponding results, provided in appendix C.5, validate the generalizability of our method to multi-reward settings while consistently achieving superior performance.

## 5.2 DB-KLA Algorithm Experiments

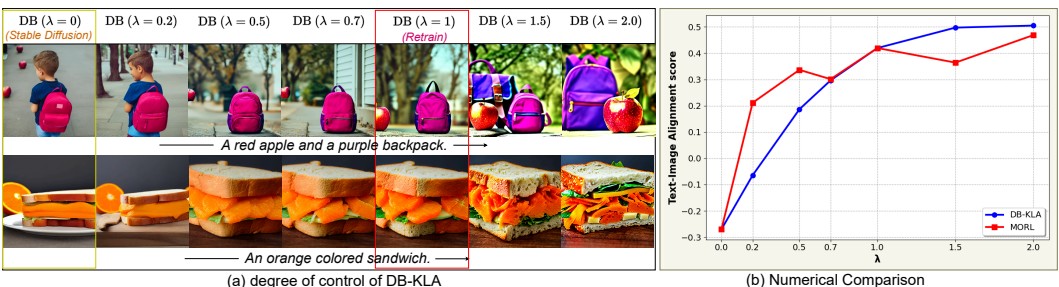

Figure 6: (a) Images generated by DB-KLA for different values of $\lambda$. We consider SDv1.5 as $\lambda = 0$ (infinite regularization weight), and as we increase $\lambda$, the aligned model moves further away from SDv1.5. These examples demonstrate that DB-KLA can smoothly control the level of text-to-image alignment by selecting different $\lambda$, without any additional fine-tuning. (b) The quantitative comparison of DB-KLA and $\lambda$-specific RL fine-tuned models among the test prompt set. Result for the train set is given in appendix D.

For the DB-KLA algorithm, we first fine-tune the baseline SVv1.5 model with the ImageReward and KL regularization weight $\alpha = 0.1$, using the *Short-DrawBench* prompts for fine-tuning and evaluation. In fig. 3(c), we show that *even without any additional fine-tuning, the images generated by DB-KLA are similar to those of $\lambda$-specific RL fine-tuned models*. In fig. 6(a), we illustrate that *DB-KLA enables smooth and continuous control over alignment strength via the rescaling factor $\lambda$*. Additional experiment results are given in appendix D. In fig. 6(b), we can observe that the average reward obtained by DB-KLA closely follows that of the MORL retrained model. As observed in fig. 6(a), in scenarios where the retrained model fails to fully align, DB-KLA with a stronger alignment setting ($\lambda > 1$) can generate more semantically accurate outputs, such as correcting object colors or preserving scene elements. This highlights its potential as a diagnostic tool for understanding and mitigating under- or over-optimization in reward-guided diffusion finetuning.

## 6 Conclusions

We introduced Diffusion Blend, a framework for inference-time multi-preference alignment in diffusion models that supports user-specified reward combinations and regularization strengths without requiring additional fine-tuning. Our proposed algorithms, DB-MPA, DB-MPA-LS, and DB-KLA, consistently outperform existing baselines and closely match the performance of individually fine-tuned models. Notably, DB-MPA-LS eliminates the linear scaling of inference time that plagues traditional inference time realignment methods, enabling efficient multi-preference alignment at scale.

## Acknowledgement

This material is based upon work partially supported by the National Science Foundation (NSF) under grants CNS-2328395, FuSe2-2425399, and NSF CAREER EPCN-2045783, the US Army Contracting Command under Contract Numbers W911NF2520046, W911NF2210151, and W911NF2120064, and the US Office of Naval Research under contract N00014-24-12615.

This research was conducted using the advanced computing resources provided by the Texas A&M High Performance Research Computing.

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

# A    TECHNICAL PROOFS

In this section, we provide the theoretical details referenced in Section 4. We begin by analyzing the solution to eq. (3), followed by a discussion of the Jensen gap approximation error introduced by interchanging the expectation operator $\mathbb{E}[\cdot]$ and the exponential function $\exp(\cdot)$. We then describe how to approximate a general function $f^{(r(w),\alpha(\lambda))}$ under the Jensen gap approximation. Finally, we provide an analysis for the sampling-based approximation algorithm based on SDE theory showing its equivalent marginal distribution.

## A.1    PROOF OF PROPOSITION 1

***Proof of Proposition 1.*** By definition, $p_t^{\mathrm{pre}}(x) = \int_y p_0^{\mathrm{pre}}(y)p_{\epsilon_t}(x - y) \ \mathrm{d}y, p_t^{\mathrm{tar}}(x) = \int_y p_0^{\mathrm{tar}}(y)p_{\epsilon_t}(x - y) \ \mathrm{d}y$. We also have, $p_{0|t}^{\mathrm{pre}}(x_0|x_t) = \frac{p_{0,t}^{\mathrm{pre}}(x_0,x_t)}{p_t^{\mathrm{pre}}(x_t)} = \frac{p_0^{\mathrm{pre}}(x_0)p_{\epsilon_t}(x_t-x_0)}{p_t^{\mathrm{pre}}(x_t)}$. Now, with an appropriate normalization constant $C$,

$$
\begin{aligned}
p_t^{\mathrm{tar}}(x) &= \int_y p_0^{\mathrm{tar}}(y)p_{\epsilon_t}(x - y) \ \mathrm{d}y = C \int_y p_0^{\mathrm{pre}}(y)\exp\left(\frac{r(y)}{\alpha}\right) p_{\epsilon_t}(x - y) \ \mathrm{d}y \\
&= Cp_t^{\mathrm{pre}}(x) \int_y \exp\left(\frac{r(y)}{\alpha}\right) \frac{p_0^{\mathrm{pre}}(y)p_{\epsilon_t}(x - y)}{p_t^{\mathrm{pre}}(x)} \ \mathrm{d}y = Cp_t^{\mathrm{pre}}(x) \int_y \exp\left(\frac{r(y)}{\alpha}\right) p_{0|t}^{\mathrm{pre}}(y|x) \ \mathrm{d}y \\
&= Cp_t^{\mathrm{pre}}(x)\mathbb{E}_{x_0\sim p_{0|t}^{\mathrm{pre}}(\cdot|x)}\left[\exp\left(\frac{r(x_0)}{\alpha}\right)\right].
\end{aligned}
$$

From this, we get,

$$
\nabla_x \log p_t^{\mathrm{tar}}(x) = \nabla_x \log p_t^{\mathrm{tar}}(x) + \nabla_x \log \mathbb{E}_{x_0\sim p_{0|t}^{\mathrm{pre}}(\cdot|x)}\left[\exp\left(\frac{r(x_0)}{\alpha}\right)\right]. \tag{11}
$$

The result now follows from the definition of $f^{(r,\alpha)}(x_t) = -\frac{1}{2}\beta(t)x_t - \beta(t)\nabla_{x_t} \log p_t^{\mathrm{tar}}(x_t)$ and $f^{\mathrm{pre}}(x_t) = -\frac{1}{2}\beta(t)x_t - \beta(t)\nabla_{x_t} \log p_t^{\mathrm{pre}}(x_t)$.    $\square$

We also present a more general formulation of proposition 1 to offer a clearer understanding.

**Proposition 3** (General statement of proposition 1). *Let $X$ be a random variable distributed according to $p_0(x)$, $Y$ be a random variable distributed as $q_0(y) = Cp_0(y)\exp(r(y)/\alpha)$, and $Z$ be an independent noise. If the probability density of $X + Z$ is $p_t$, then the probability density of $Y + Z$ is $q_t(x) = Cp_t(x)\mathbb{E}[\exp(\frac{r(X)}{\alpha})|X + Z = x]$. The score of $Y + Z$ is given by $\nabla_x \log \mathbb{E}[\exp(\frac{r(X)}{\alpha})|X + Z = x] + \nabla \log p_t(x)$.*

*Proof.* Note $X + Z$ is distributed w.r.t. the probability density:

$$
p_t(x) = \int p_0(y)p_Z(x - y)dy.
$$

Density of $Y + Z$ is:

$$
\begin{aligned}
q_t(x) \\
&= \int q_0(y)p_Z(x - y)dy \\
&= C \int p_0(y)\exp(\frac{r(y)}{\alpha})p_Z(x - y)dy \\
&= Cp_t(x) \int \exp(\frac{r(y)}{\alpha})\frac{p_0(y)p_Z(x - y)}{\int p_0(s)p_Z(x - s)ds}dy \\
&= Cp_t(x) \int \exp(\frac{r(y)}{\alpha})p_{X|X+Z=x}(X = y)dy \\
&= Cp_t(x)\mathbb{E}\left[\exp(\frac{r(X)}{\alpha})\Big|X + Z = x\right].
\end{aligned}
$$

$\square$

*Remark* 4. Consider the variance-exploding forward process proposed by Song et al. (2021), i.e. $X_t = X_0 + \sqrt{\bar{\alpha}_t} Z$ with $Z \sim \mathcal{N}(0, I)$. proposition 3 implies that for two different initial distributions $X \sim p_0$ and $Y \sim q_0 \propto p_0 \exp(r/\alpha)$, both following the same noise-exploding forward process, the marginal probability density $q_t(y)$ of $Y_t = Y_0 + \sqrt{\bar{\alpha}_t} Z$ is equal to the product of the density of $X_t$ and a posterior mean term $\mathbb{E}[\exp(r(X_0)/\alpha) \mid X_t = y]$. Recall that the reverse process involves adding an extra score term $\nabla_x \log p_t(x)$ into the drift. proposition 3 suggests that approximating $p_0^{\text{pre}} \exp(r/\alpha)$ in the reverse process can be achieved by replacing the original score $\nabla \log p_t(x)$ in $X_t$, with a shifted score $\nabla_x \log p_t(x) + \nabla_x \log \mathbb{E}[\exp(r(X_0)/\alpha) \mid X_t = x]$.

The same analysis applies to the variance-preserving forward process used in DDPM (Ho et al., 2020), defined as $X_t = \sqrt{\bar{\alpha}_t} X_0 + \sqrt{1 - \bar{\alpha}_t} Z$ with $Z \sim \mathcal{N}(0, I)$. In this case, the same conclusion follows by replacing the random variable $X$ in proposition 3 with $\sqrt{\bar{\alpha}_t} X_0$.

## A.2   APPROXIMATION ERROR UPPER BOUND FOR EQ. (8)

**Lemma 3.** *Decompose the reward function into two parts*

$$\frac{r(x_0^{\text{pre}})}{\alpha} = \tilde{r}(x_t^{\text{pre}}, t) + \eta(\omega, x_t^{\text{pre}}, t)$$

*where $\tilde{r}(x_t^{\text{pre}}, t) = \mathbb{E}[\frac{r(x_0^{\text{pre}})}{\alpha} | x_t^{\text{pre}}]$ only depends on $x_t^{\text{pre}}$ and $\eta(\omega, x_t^{\text{pre}}, t) = \frac{r(x_0^{\text{pre}})}{\alpha} - \tilde{r}(x_t^{\text{pre}}, t)$ contains randomness $\omega$ induced from the noise injection process. Let $p_{R|t}$ denote the conditional distribution of random variable $R := r(x_0^{\text{pre}})/\alpha$ given $x_t^{\text{pre}}$. Then,*

$$\|\nabla_x \log \mathbb{E}[\exp(r(x_0^{\text{pre}})/\alpha)|x_t^{\text{pre}} = x] - \nabla_x \mathbb{E}[r(x_0^{\text{pre}})/\alpha|x_t^{\text{pre}} = x]\| \leq L_{t,1}(x) \times L_{t,2}(x) + L_{t,3}(x),$$

*where*

$$L_{t,1}(x) = \sqrt{\mathbb{E}[\|\nabla_x \eta(\omega, x_t^{\text{pre}}, t)\|_2^2 | x_t^{\text{pre}} = x]},$$

$$L_{t,2}(x) = \frac{\sqrt{\text{Var}(\exp(r(x_0^{\text{pre}})/\alpha)|x_t^{\text{pre}} = x)}}{\mathbb{E}[\exp(r(x_0^{\text{pre}})/\alpha)|x_t^{\text{pre}} = x]},$$

$$L_{t,3}(x) = (1 + \frac{1}{\alpha}) \sup_r \|\nabla_x \log p_{R|t}(r|x_t^{\text{pre}} = x) + \nabla_r \log p_{R|t}(r|x_t^{\text{pre}} = x)\|.$$

*Proof.* To shorten the notation, we denote random variable $R := r(x_0^{\text{pre}})/\alpha$ and constant $C_R = \max_x |r(x)/\alpha| = \frac{1}{\alpha}$. Let $p_{R|t}$ be the conditional probability of $R$ given $x_t$.

We first make one assumption about the boundary condition that the conditional probability density decreases exponentially fast $\lim_{r \to \pm\infty} r \cdot p_{R|t}(r|x) = 0$, and $\lim_{r \to \pm\infty} e^r \cdot p_{R|t}(r|x) = 0$. It is known that sub-Gaussian distributions satisfy those assumptions with an exponentially decreasing tail $p(x) \leq C e^{-|x|^a}$ for large enough $x$. Remember that in our experimental setting, all reward models output in a bounded range $[-2, 2]$ and $\alpha$ is a fixed real number. Therefore, for the bounded random variable $R$, it belongs to the sub-Gaussian distribution and satisfies our boundary assumption.

Let $F(x) = \mathbb{E}[\exp(R)|x_t^{\text{pre}} = x]$ and $g(x) = \mathbb{E}[R|x_t^{\text{pre}} = x]$, then

$$\nabla_x \log F(x) - \nabla_x g(x) = \frac{\nabla_x F(x)}{F(x)} - \nabla_x g(x)$$

$$= \frac{\int \nabla_x \exp(r) \cdot p_{R|t}(r|x) dr + \int \exp(r) \nabla_x p_{R|t}(r|x) dr}{F(x)}$$

$$- \left( \int \nabla_x r \cdot p_{R|t}(r|x) + r \nabla_x p_{R|t}(r|x) dr \right)$$

$$= \frac{\mathbb{E}[\nabla_{x_t^{\text{pre}}} \exp(R)|x_t^{\text{pre}} = x]}{F(x)} - \mathbb{E}[\nabla_{x_t^{\text{pre}}} R|x_t^{\text{pre}} = x]$$

$$+ \left( \frac{\int \exp(r) \nabla_x p_{R|t}(r|x) dr}{F(x)} - \int r \nabla_x p_{R|t}(r|x) dr \right)$$

$$= \underbrace{\mathbb{E}\left[ \left( \frac{\exp(R)}{F(x)} - 1 \right) \nabla_{x_t^{\text{pre}}} R \Big| x_t^{\text{pre}} = x \right]}_{I_1} + \underbrace{\left( \frac{\int \exp(r) \nabla_x p_{R|t}(r|x) dr}{F(x)} - \int r \nabla_x p_{R|t}(r|x) dr \right)}_{I_2}.$$

We first bound the first term $I_1$. Decompose $R$ into two parts $R = \tilde{r}(x_t^{\mathrm{pre}}, t) + \eta(\omega, x_t^{\mathrm{pre}}, t)$ where the first part $\tilde{r}(x_t^{\mathrm{pre}}, t)$ only depends on $x_t^{\mathrm{pre}}$. Note that $\mathbb{E}[\frac{\exp(R)}{F(x)} - 1|x_t^{\mathrm{pre}} = x] = \mathbb{E}[\frac{\exp(R)}{\mathbb{E}[\exp(R)|x_t^{\mathrm{pre}}=x]} - 1|x_t^{\mathrm{pre}} = x] = 0$.

$$
\begin{aligned}
\|I_1\| &= \left\| \mathbb{E}\left[ \left( \frac{e^R}{F(x)} - 1 \right) \nabla_{x_t^{\mathrm{pre}}} R \middle| x_t^{\mathrm{pre}} = x \right] \right\| \\
&= \left\| \mathbb{E}\left[ \left( \frac{e^R}{F(x)} - 1 \right) (\nabla_{x_t^{\mathrm{pre}}} \tilde{r}(x_t^{\mathrm{pre}}, t)) + \nabla_{x_t^{\mathrm{pre}}} \eta(\omega, x_t^{\mathrm{pre}}, t)) \middle| x_t^{\mathrm{pre}} = x \right] \right\| \\
&= \left\| \mathbb{E}\left[ \left( \frac{e^R}{F(x)} - 1 \right) \nabla_{x_t^{\mathrm{pre}}} \eta(\omega, x_t^{\mathrm{pre}}, t) \middle| x_t^{\mathrm{pre}} = x \right] \right\| \\
&\leq \sqrt{ \mathbb{E}\left[ \left( \frac{e^R}{F(x)} - 1 \right)^2 \middle| x_t^{\mathrm{pre}} = x \right] } \times \sqrt{ \mathbb{E}\left[ \|\nabla_{x_t^{\mathrm{pre}}} \eta(\omega, x_t^{\mathrm{pre}}, t)\|_2^2 \middle| x_t^{\mathrm{pre}} = x \right] } \\
&= L_{t,2}(x) \times L_{t,1}(x).
\end{aligned}
$$

For the second term $I_2$, we note that $I_2 \equiv 0$ under the assumption proposed by Uehara et al. (2024b) that $R = f(x_t) + \epsilon$ can be decomposed to the summation of a function related to $x_t$ and an independent noise $\epsilon$, which is induced by the translation invariance of $p_{R|t}(r|x) = p_{\mathrm{noise}}(r - x)$. Inspired by this observation, we define $\Delta_t := \nabla_x \log p_{R|t}(r|x) + \nabla_r \log p_{R|t}(r|x)$ to measure the shift from such a translation invariant family $\{p_{R|t} : \exists\, p_{noise}, \text{ s.t. } p_{R|t}(r|x) = p_{\mathrm{noise}}(r - x)\}$.

$$
\begin{aligned}
I_2 &= \frac{\int \exp(r) p_{R|t}(r|x) \nabla_x \log p_{R|t}(r|x) dr}{F(x)} - \int r p_{R|t}(r|x) \nabla_x \log p_{R|t}(r|x) dr \\
&= \frac{\int \exp(r) p_{R|t}(r|x) (\Delta_t - \nabla_r \log p_{R|t}(r|x)) dr}{F(x)} - \int r p_{R|t}(r|x) (\Delta_t - \nabla_r \log p_{R|t}(r|x)) dr \\
&= \mathbb{E}_{p'}[\Delta_t | x_t^{\mathrm{pre}} = x] - \mathbb{E}[r \Delta_t | x_t^{\mathrm{pre}} = x] \\
&\quad - \Big( \underbrace{\frac{\int \exp(r) p_{R|t}(r|x) \nabla_r \log p_{R|t}(r|x) dr}{F(x)}}_{I_F} - \underbrace{\int r p_{R|t}(r|x) \nabla_r \log p_{R|t}(r|x) dr}_{I_g} \Big),
\end{aligned}
$$

where $p'(r|x) = \frac{\exp(r) p_{R|t}(r|x)}{F(x)}$ is the reweighted probability. Under the boundary condition, we can show that $I_F = I_g = -1$ as

$$
\begin{aligned}
I_g &= \int r \nabla_r p_{R|t}(r|x) dr = \int \nabla_r (r p_{R|t}(r|x)) dr - \int p_{R|t}(r|x) dr \\
&= r p_{R|t}(r|x) \Big|_{r=-\infty}^{r=+\infty} - 1 = -1, \\
I_F &= \frac{\int \exp(r) \nabla_r p_{R|t}(r|x) dr}{F(x)} = \frac{\int \nabla_r \big( \exp(r) p_{R|t}(r|x) \big) dr - \int \exp(r) p_{R|t}(r|x) dr}{F(x)} \\
&= \frac{\int \nabla_r (\exp(r) p_{R|t}(r|x)) dr}{F(x)} - 1 = \frac{\big( \exp(r) p_{R|t}(r|x) \big) \Big|_{r=-\infty}^{r=+\infty}}{F(x)} - 1 = -1.
\end{aligned}
$$

Therefore, $\|I_2\| = \|\mathbb{E}_{p'}[\Delta_t | x_t^{\mathrm{pre}} = x] - \mathbb{E}[r \Delta_t | x_t^{\mathrm{pre}} = x]\| \leq (1 + C_R) \sup_{r,x} \|\Delta_t\| = L_{t,3}(x)$. $\quad\square$

## A.3 PROOF OF LEMMA 2

***Proof of Lemma 2.*** By leveraging the linearity of expectation available in $\bar{u}^{(r(w),\alpha)}$, we get

$$
\bar{u}^{(r(w),\alpha)}(x, t) = \nabla_x \mathbb{E}_{x_0 \sim p_{0|t}^{\mathrm{pre}}(\cdot|x)} \left[ \left( \frac{\sum_{i=1}^m w_i r_i(x_0)}{\alpha} \right) \right] = \sum_{i=1}^m w_i \bar{u}^{(r_i,\alpha)}(x, t).
$$

Using this, we get

$$
\begin{aligned}
&f^{(r(w),\alpha)}(x_t,t)\\
=&f^{\mathrm{pre}}(x_t,t)-\beta(t)u^{(r(w),\alpha)}(x_t,t)=f^{\mathrm{pre}}(x_t,t)-\beta(t)\bar{u}^{(r(w),\alpha)}(x,t)-\beta(t)\Delta^{(r(w),\alpha)}(x,t)\\
=&f^{\mathrm{pre}}(x_t,t)-\beta(t)\sum_{i=1}^m w_i u^{(r_i,\alpha)}(x,t)+\beta(t)\left(\sum_{i=1}^m w_i\Delta^{(r_i,\alpha)}(x,t)-\Delta^{(r(w),\alpha)}(x,t)\right)\\
=&\sum_{i=1}^m w_i(f^{\mathrm{pre}}(x_t,t)-\beta(t)u^{(r_i,\alpha)}(x,t))+\beta(t)\left(\sum_{i=1}^m w_i\Delta^{(r_i,\alpha)}(x,t)-\Delta^{(r(w),\alpha)}(x,t)\right)\\
=&\sum_{i=1}^m w_i f^{(r_i,\alpha)}(x_t,t)+\beta(t)\left(\sum_{i=1}^m w_i\Delta^{(r_i,\alpha)}(x,t)-\Delta^{(r(w),\alpha)}(x,t)\right).
\end{aligned}
$$

Similarly, using the fact that $\bar{u}^{(r,\alpha(\lambda))}(x,t)=\lambda\bar{u}^{(r,\alpha)}(x,t)$, we get

$$
\begin{aligned}
&f^{(r,\alpha(\lambda))}(x_t,t)\\
=&f^{\mathrm{pre}}(x_t,t)-\beta(t)u^{(r,\alpha(\lambda))}(x_t,t)=f^{\mathrm{pre}}(x_t,t)-\beta(t)\bar{u}^{(r,\alpha(\lambda))}(x,t)-\beta(t)\Delta^{(r,\alpha(\lambda))}(x,t)\\
=&f^{\mathrm{pre}}(x_t,t)-\beta(t)\lambda\bar{u}^{(r,\alpha)}(x,t)-\beta(t)\Delta^{(r,\alpha(\lambda))}(x,t)\\
=&\lambda(f^{\mathrm{pre}}(x_t,t)-\beta(t)\bar{u}^{(r,\alpha)}(x,t))+(1-\lambda)f^{\mathrm{pre}}(x_t,t)-\beta(t)\Delta^{(r,\alpha(\lambda))}(x,t)\\
=&\lambda f^{(r,\alpha)}(x_t,t)+(1-\lambda)f^{\mathrm{pre}}(x_t,t)+\beta(t)\left(\lambda\Delta^{(r,\alpha)}(x,t)-\Delta^{(r,\alpha(\lambda))}(x,t)\right).
\end{aligned}
$$

$\square$

### A.4  PROOF OF PROPOSITION 2

***Proof of Proposition 2.*** Denote $b(x)=af_1(x)+(1-a)f_2(x)$. For any function $\psi\in C^2(\mathbb{R}^d)$, Itô's formula gives:

$$
d\psi(X_t^1)=\sigma(t)\nabla\psi(X_t^1)^T d\omega_t+[\nabla\psi(X_t^1)^T b(X_t^1)+\frac{\sigma^2(t)}{2}tr(\nabla^2\psi(X_t^1))]dt,
$$

$$
\frac{d}{dt}\mathbb{E}[\psi(X_t^1)]=\int\sigma(t)\nabla\psi(X_t^1)^T d\omega_t+\mathbb{E}[\nabla\psi(X_t^1)^T b(X_t^1)+\frac{\sigma^2(t)}{2}tr(\nabla^2\psi(X_t^1))].
$$

Note that $tr(\nabla^2\psi)=\Delta\psi$, and $\int\sigma(t)\nabla\psi(X_t^1)^T d\omega_t$ is a martingale with mean 0, therefore the first term in the RHS is 0. We got for SDE 1:

$$
\frac{d}{dt}\mathbb{E}[\psi(X_t^1)]=\mathbb{E}[\nabla\psi(X_t^1)^T b(X_t^1)+\frac{\sigma^2(t)}{2}\Delta\psi(X_t^1)].
$$

Move to the SDE 2. Similarly, we can derive:

$$
\begin{aligned}
\frac{d}{dt}\mathbb{E}[\psi(X_t^2)]&=\mathbb{E}[\nabla\psi(X_t^2)^T(Y_t f_1(X_t^2)+(1-Y_t)f_2(X_t^2))+\frac{\sigma^2(t)}{2}\Delta\psi(X_t^2)]\\
&=\mathbb{E}[Y_t\nabla\psi(X_t^2)^T f_1(X_t^2)]+\mathbb{E}[(1-Y_t)\nabla\psi(X_t^2)^T f_2(X_t^2)]+\mathbb{E}[\frac{\sigma^2(t)}{2}\Delta\psi(X_t^2)]\\
&=\mathbb{E}[Y_t]\mathbb{E}[\nabla\psi(X_t^2)^T f_1(X_t^2)]+\mathbb{E}[1-Y_t]\mathbb{E}[\nabla\psi(X_t^2)^T f_2(X_t^2)]+\mathbb{E}[\frac{\sigma^2(t)}{2}\Delta\psi(X_t^2)]\\
&=a\mathbb{E}[\nabla\psi(X_t^2)^T f_1(X_t^2)]+(1-a)\mathbb{E}[\nabla\psi(X_t^2)^T f_2(X_t^2)]+\mathbb{E}[\frac{\sigma^2(t)}{2}\Delta\psi(X_t^2)]\\
&=\mathbb{E}[\nabla\psi(X_t^2)b(X_t^2)+\frac{\sigma^2(t)}{2}\Delta\psi(X_t^2)],
\end{aligned}
$$

where the independence of $\{Y_t\}$ is applied.

Both $\{X_t^1\}$ and $\{X_t^2\}$ satisfy: $\frac{d}{dt}\mathbb{E}[\psi(X_t)] = \mathbb{E}[\nabla\psi(X_t)b(X_t) + \frac{\sigma^2(t)}{2}\Delta\psi(X_t)]$ with $X_0 \sim p_0$, for any $\psi \in C^2$. Denote $p_t$ as the probability density of $X_t$, we got:

$$\frac{d}{dt}\int \psi(s)p_t(x)dx = \int \left(\nabla\psi(x)b(x) + \frac{\sigma^2(t)}{2}\Delta\psi(x)\right)p_t(x)dx$$

$$\int \psi(s)\partial_t p_t(x)dx = \int -\psi(x)\nabla\Big(b(x)p_t(x)\Big) + \frac{\sigma^2(t)}{2}\psi(x)\Delta p_t(x)dx,$$

where the change of integral is used. Then we have:

$$\int \psi(s)\Big(\partial_t p_t(x)dx + \nabla(b(x)p_t(x)) - \frac{\sigma^2(t)}{2}\Delta p_t(x)\Big)dx = 0, \ \forall\psi \in C^2.$$

Therefore, both $p_{X_t^1}$ and $p_{X_t^2}$ are solutions of this PDE: $\partial_t p_t(x)dx + \nabla(b(x)p_t(x)) - \frac{\sigma^2(t)}{2}\Delta p_t(x) = 0$ with $p|_{t=0} = p_0$. Since the drift term $b$ is Lipschitz continuous, this PDE has only one solution. Therefore, $p_{X_t^1} = p_{X_t^2}$. $\qquad\square$

## B  EXPERIMENTAL DETAILS

### B.1  BASELINE ALGORITHMS

**MORL:** For MORL, we fine-tune the SDv1.5 base model using RL, following the same procedure as in Fan et al. (2023). In particular, we obtain separately fine-tuned models for $(r(w), \alpha(\lambda))$, for different values of $w$ and $\lambda$. This is used as an oracle baseline for both DB-MPA and DB-KLA.

**Rewarded Soup (RS):** We use RS, introduced by Rame et al. (2023), as a baseline for the DB-MPA. We first RL fine-tune SDv1.5 separately for each reward each reward $(r_i)_{i=1}^m$ with a fixed $\alpha$. So, starting from the pre-trained model parameter $\theta^{\text{pre}}$, we obtained $m$ RL fine-tuned models with parameters $(\theta_i^{\text{rl}})_{i=1}^m$. At inference time, given the preference $w$ given by the user, we construct a new model with parameter $\theta^{\text{rs}}(w) = \sum_{i=1}^m w_i\theta_i^{\text{rl}}$, and generate images using this model. We only average the U-Net parameters from the fine-tuned models.

**Reward Gradient Guidance (RGG):** We follow the gradient guidance approach (Chung et al., 2023; Kim et al., 2025) where the diffusion process at each backward step is updated using the gradient of the reward function. The update rule is given by:

$$\mu_\theta(x_t, t) + \frac{\lambda_{t-1}\sigma_t^2}{\alpha}\nabla_{x_t}\hat{r}(x_t), \tag{12}$$

where $\mu_\theta(x_t, t)$ denotes the base model's predicted mean, $\sigma_t^2$ is the noise variance at timestep $t$, $\alpha$ is the KL regularization weight, and $\lambda_{t-1}$ is a time-dependent scaling factor defined by the exponential schedule $\lambda_t = (1 + \gamma)^{t-1}$ with $\gamma = 0.024$, as introduced in Kim et al. (2025). The reward function is defined as $\hat{r}(x_t) = r(x_0(x_t))$, where $x_0(x_t)$ denotes the Tweedie approximation-based reconstructed image obtained from $x_t$ using the denoiser network.

We adapted this approach to the multi-objective setting by combining gradients from two reward functions $r_1$ and $r_2$ as,

$$\nabla_{x_t}\hat{r}(x_t) = w_1\nabla_{x_t}r_1(x_0(x_t)) + (1 - w_1)\nabla_{x_t}r_2(x_0(x_t)). \tag{13}$$

However, to ensure that the influence of each reward is independent of its scale or gradient magnitude, we normalized the individual gradients before combining them,

$$\nabla_{x_t}\hat{r}(x_t) = w_1\frac{\nabla_{x_t}r_1(x_0(x_t))}{\|\nabla_{x_t}r_1(x_0(x_t))\|} + (1 - w_1)\frac{\nabla_{x_t}r_2(x_0(x_t))}{\|\nabla_{x_t}r_2(x_0(x_t))\|}. \tag{14}$$

This normalization ensures that the reward guidance strength is controllable and not biased by the nature or scale of individual rewards.

**CoDe:** Introduced by Singh et al. (2025), CoDe is a gradient-free guidance method for aligning diffusion models with downstream reward functions. CoDe operates by partitioning the denoising process into blocks and, at each block, generating multiple candidate samples. It then selects the sample with the highest estimated lookahead reward to proceed with the next denoising steps. For our experiments, we configured CoDe with 20 particles and a lookahead of 5 steps.

## B.2 PSEUDO CODE

In this section, we present the pseudo-code for the three inference-time algorithms introduced in section 4: DB-MPA, DB-KL, DB-MPA-LS. All algorithms leverage the approximation result from lemma 2 or proposition 2 to enable controllable sampling without requiring additional fine-tuning at inference time.

Algorithm 1 outlines the DB-MPA procedure, in which a user-specified preference vector $w$ is used to linearly combine the drift functions of $m$ RL fine-tuned models, each optimized independently for a distinct reward basis. Algorithm 2 presents the DB-KLA procedure, which instead blends the drift of a fine-tuned model with the pre-trained model. Algorithm 3 approximates Algorithm 1 by sampling a drift with the assigned preference weights. All fine-tuned models used in algorithms are obtained by applying the single-reward RL fine-tuning algorithm individually to each reward function.

Note that all algorithms adopt a basic Euler-Maruyama discretization to simulate the reverse SDE in eq. (5). In practice, this integration step can be replaced by any diffusion model's reverse process (e.g., DDIM (Song et al., 2020) or other solvers), as long as the drift term (or the predicted denoising output) is appropriately mixed.

---

**Algorithm 1 DB-MPA**

---

**Input:** RL-fine-tuned drifts $\{ f^{(r_i,\alpha)} \}_{i=1}^m$; weights $w \in \mathbb{R}^m$, $\sum_{i=1}^m w_i = 1$; time grid $0 = t_0 < t_1 < \cdots < t_N = T$
1: Sample $x_{t_N} \sim \mathcal{N}(0, I)$
2: **for** $k \leftarrow N$ **down to** 1 **do**
3:      $\Delta t_k \leftarrow t_k - t_{k-1}$                                        ▷ positive
4:      noise $z \sim \mathcal{N}(0, I)$
5:      $f_{\text{mix}} \leftarrow \sum_{i=1}^m w_i \, f^{(r_i,\alpha)}(x_{t_k}, t_k)$
6:      $x_{t_{k-1}} \leftarrow x_{t_k} - f_{\text{mix}} \, \Delta t_k + \sigma(t_k)\sqrt{\Delta t_k} \, z$
7: **end for**
**Output:** $x_{t_0}$

---

---

**Algorithm 2 DB-KLA**

---

**Input:** RL-fine-tuned drift $f^{(r,\alpha)}$; original pretrained drift $f^{pre}$; KL-reweight parameter $\lambda \geq 0$; time grid $0 = t_0 < t_1 < \cdots < t_N = T$
1: Sample $x_{t_N} \sim \mathcal{N}(0, I)$
2: **for** $k \leftarrow N$ **down to** 1 **do**
3:      $\Delta t_k \leftarrow t_k - t_{k-1}$                                        ▷ positive
4:      noise $z \sim \mathcal{N}(0, I)$
5:      $f_{\text{KL}} \leftarrow \lambda f_{t_k}^{(r,\alpha)}(x_{t_k}, t_k) + (1 - \lambda) f_{t_k}^{pre}(x_{t_k}, t_k)$
6:      $x_{t_{k-1}} \leftarrow x_{t_k} - f_{\text{KL}} \, \Delta t_k + \sigma(t_k)\sqrt{\Delta t_k} \, z$
7: **end for**
**Output:** $x_{t_0}$

---

## B.3 PROMPT SETS

We evaluate our methods on two datasets: *Short-DrawBench* and *GenEval*.

**Short-DrawBench.** The original DrawBench dataset (Saharia et al., 2022b) contains 183 prompts across 11 categories. For our experiments, we isolate the 25 prompts in the `color` category, forming the *Short-DrawBench* subset. This reduced scale enables direct comparison with MORL, since training separate models for each preference or KL weight is computationally feasible. To test generalization, we further construct a set of 1,000 evaluation prompts generated by GPT-4 (Achiam et al., 2023). These prompts introduce novel object–color and multi-object compositions not present in the training subset. The instruction used for generating this test set was:

---

**Algorithm 3 DB-MPA-LS**

---

**Input:** RL-fine-tuned drifts $\{f^{(r_i,\alpha)}\}_{i=1}^m$; weights $w \in \mathbb{R}^m$, $\sum_{i=1}^m w_i = 1$; time grid $0 = t_0 < t_1 < \cdots < t_N = T$

1: Sample $x_{t_N} \sim \mathcal{N}(0, I)$
2: **for** $k \leftarrow N$ **down to** 1 **do**
3:      $\Delta t_k \leftarrow t_k - t_{k-1}$                                                    $\triangleright$ positive
4:      noise $z \sim \mathcal{N}(0, I)$
5:      sample $i_k$ from $\{1, 2, \ldots, m\}$ with probability $(w_1, w_2, \ldots, w_m)$
6:      $x_{t_{k-1}} \leftarrow x_{t_k} - f^{(r_{i_k},\alpha)}(x_{t_k}, t_k)\,\Delta t_k + \sigma(t_k)\sqrt{\Delta t_k}\,z$
7: **end for**

**Output:** $x_{t_0}$

---

*"Please generate 1000 testing prompts that are similar to the following training prompts, which are color+object combinations. You should use colors that appeared in the train set or have a similar semantic meaning. Objects can be a little more common or random."*

**GenEval.** The GenEval benchmark (Kirstain et al., 2023) consists of 550 prompts designed to test compositional generalization, spanning attributes such as color, counting, spatial relations, and multi-object scenes. In addition to the official prompts, we generate an extra 700 held-out evaluation prompts using the official GenEval prompt generation toolkit. These follow the same construction rules as the original dataset, obtained by varying random seeds and object/color assignments.

### B.4 COMPUTING HARDWARE AND HYPERPARAMETERS

Fine-tuning of Stable Diffusion for each KL weight and reward composition was performed on NVIDIA A100 GPUs using mixed precision. We used the AdamW optimizer with a learning rate of $1 \times 10^{-5}$ for policy updates and applied LoRA with rank 4. Gradient accumulation was set to 12, with a per-GPU batch size of 2 for policy updates and 6 for prompt sampling.

Policy updates followed a clipped PPO-style objective with a clipping ratio of $1 \times 10^{-4}$ following Fan et al. (2023). Each outer iteration performed 5 policy gradient steps and 5 value function updates (batch size 256, learning rate $1 \times 10^{-4}$), using a replay buffer of size 1000. Training required approximately 96,000 online samples to converge for the *Short-DrawBench* subset. A similar size of online samples is used for GenEval.

## C  DB-MPA ALGORITHM: ADDITIONAL RESULTS

In this section, we present additional experimental results for the DB-MPA algorithm. We begin with reward evaluations on the training prompts. Next, we demonstrate that DB-MPA naturally extends to alignment with three rewards ($m = 3$). We also show that DB-MPA-LS produces outputs visually close to DB-MPA. We then provide additional qualitative comparisons with baselines to further highlight DB-MPA's effectiveness. Finally, we evaluate DB-MPA's ability to achieve fine-grained multi-preference alignment.

### C.1  RESULTS ON TRAINING PROMPTS

For the *Short-DrawBench* setting, Table 2 reports the performance of DB-MPA and baseline algorithms relative to Stable Diffusion, evaluated on the training prompts (30 random seed per prompt, 750 images in total). Across all preference weights, DB-MPA consistently outperforms RS, CoDe, and RGG.

We next consider the *GenEval* dataset, which evaluates compositional generalization. Table 3 reports the reward improvements of all methods relative to Stable Diffusion. DB-MPA and DB-MPA-LS achieve the best or near-best gains across most preference weights. Although the table lists separate $\Delta r_1$ and $\Delta r_2$, we also computed the weighted reward $\Delta\mathrm{WR} = wr_1 + (1-w)r_2$. On this aggregate metric, DB-MPA and DB-MPA-LS consistently outperform all other baselines, confirming their superior trade-off performance.

Table 2: Quantitative comparison of DB-MPA and baseline methods on train prompts. Here $\Delta r_i = r_i - r_i^{\text{SD}}$

|  |  | DB-MPA | | RS | | CoDe | | RGG | |
|---|---|---|---|---|---|---|---|---|---|
|  |  | $\Delta r_1$ | $\Delta r_2$ | $\Delta r_1$ | $\Delta r_2$ | $\Delta r_1$ | $\Delta r_2$ | $\Delta r_1$ | $\Delta r_2$ |
|  | $w$=0.2 | **0.19** | **0.74** | -0.01 | 0.61 | 0.14 | 0.23 | 0.12 | 0.59 |
| Improvement (↑) | $w$=0.5 | **0.49** | **0.50** | 0.12 | 0.20 | 0.29 | 0.19 | 0.13 | 0.39 |
|  | $w$=0.8 | **0.65** | **0.18** | 0.54 | 0.02 | 0.34 | 0.12 | 0.03 | 0.16 |

Table 3: GenEval train results: reward improvements ($\Delta r$) relative to Stable Diffusion.

|  |  | DB-MPA | | DB-MPA-LS | | RS | | CoDe | | RGG | |
|---|---|---|---|---|---|---|---|---|---|---|---|
|  |  | $\Delta r_1$ | $\Delta r_2$ | $\Delta r_1$ | $\Delta r_2$ | $\Delta r_1$ | $\Delta r_2$ | $\Delta r_1$ | $\Delta r_2$ | $\Delta r_1$ | $\Delta r_2$ |
|  | $w$=0.2 | +0.122 | **+0.497** | +0.143 | +0.477 | -0.021 | +0.378 | **+0.254** | +0.107 | +0.064 | +0.497 |
|  | $w$=0.5 | +0.267 | **+0.357** | +0.376 | +0.310 | +0.161 | +0.192 | +0.344 | +0.067 | +0.104 | +0.327 |
| Reward (↑) | $w$=0.8 | +0.382 | **+0.185** | +0.411 | +0.180 | +0.340 | +0.089 | +0.374 | +0.007 | -0.246 | +0.111 |

## C.2 QUANTITATIVE RESULTS ON GENEVAL TEST DATA

Table 4 reports the raw numerical values corresponding to fig. 5 shown in the main paper.

Table 4: GenEval (test): numerical results ($r_1$, $r_2$) corresponding to the Pareto-front plot fig. 5 in the main text. DB-MPA and DB-MPA-LS consistently dominate baselines across preference weights.

|  |  | SD | | DB-MPA | | DB-MPA-LS | | RS | | CoDe | | RGG | |
|---|---|---|---|---|---|---|---|---|---|---|---|---|---|
|  |  | $r_1$ | $r_2$ | $r_1$ | $r_2$ | $r_1$ | $r_2$ | $r_1$ | $r_2$ | $r_1$ | $r_2$ | $r_1$ | $r_2$ |
|  | $w$=0 | -0.2 | -0.04 | -0.19 | 0.83 | -0.19 | 0.83 | -0.19 | 0.83 | -0.03 | 0.07 | 0.03 | 0.52 |
|  | $w$=0.2 | -0.2 | -0.04 | -0.05 | 0.73 | -0.07 | 0.69 | -0.21 | 0.54 | -0.07 | 0.05 | 0.02 | 0.44 |
| Reward (↑) | $w$=0.5 | -0.2 | -0.04 | 0.13 | 0.47 | 0.19 | 0.37 | -0.09 | 0.21 | 0.05 | 0.01 | -0.35 | 0.27 |
|  | $w$=0.8 | -0.2 | -0.04 | 0.25 | 0.18 | 0.25 | 0.13 | 0.12 | 0.07 | 0.12 | -0.05 | -0.41 | 0.10 |
|  | $w$=1 | -0.2 | -0.04 | 0.26 | 0.01 | 0.26 | 0.01 | 0.26 | 0.01 | 0.12 | -0.09 | -0.20 | -0.06 |

## C.3 RESULTS FOR CONFLICTING REWARDS

To evaluate DB-MPA under adversarial objectives, we consider the conflict between *JPEG compressibility* and *VILA aesthetics*. The JPEG reward incentivizes smooth, low-detail images, whereas VILA prioritizes fine-grained, high-quality visuals. These objectives are naturally at odds: optimizing for JPEG typically harms aesthetics. For example, while the Stable Diffusion (SD) baseline scores $r_1 = -0.09$ on JPEG and $r_2 = 0.48$ on VILA, an RL-fine-tuned JPEG model attains $r_1 = 1.52$ but drops to $r_2 = -0.40$.

We train a JPEG-aligned model ($r_1$) and combine it with our VILA-aligned model ($r_2$) using DB-MPA. Because JPEG compressibility is non-differentiable, gradient-based methods such as RGG cannot be applied. We therefore compare DB-MPA against Rewarded Soup (RS) and CoDe. Rewards are reported as a function of the blending weight $w \in [0, 1]$, with $w = 1$ preferring JPEG and $w = 0$ preferring VILA.

Across all weights, DB-MPA achieves substantially higher weighted rewards than both RS and CoDe. This demonstrates that DB-MPA can effectively balance two strongly conflicting objectives far better than competing baselines.

## C.4 EFFECT OF INCREASING THE NUMBER OF REWARDS

We study how the performance of DB changes as the number of reward models increases. In Figure fig. 7, we evaluate DB under 2-, 3-, and 4-reward settings and compare DB-MPA, DB-MPA-LS, and RS, all of which interpolate the same set of finetuned reward-basis models. It can be observed that the improvements of DB-MPA and DB-MPA-LS over the pretrained model remain stable as

Table 5: Performance on Short-drawbench 1k test prompts under conflicting rewards: JPEG compressibility ($r_1$) and VILA aesthetics ($r_2$). We also report the weighted reward $WR = wr_1 + (1 - w)r_2$. The best weighted reward is bold.

|  |  | DB-MPA | | | RS | | | CoDe | | |
|---|---|---|---|---|---|---|---|---|---|---|
|  |  | $r_1$ | $r_2$ | WR | $r_1$ | $r_2$ | WR | $r_1$ | $r_2$ | WR |
| Reward (↑) | $w$=0.2 | 0.52 | 0.40 | **0.44** | 0.11 | 0.28 | 0.21 | 0.04 | 0.02 | 0.03 |
|  | $w$=0.5 | 1.00 | 0.18 | **0.59** | 0.30 | 0.02 | 0.16 | 0.22 | 0.03 | 0.12 |
|  | $w$=0.8 | 1.35 | -0.18 | **0.88** | 0.93 | -0.13 | 0.72 | 0.37 | -0.01 | 0.29 |

more rewards are introduced. All experiments use uniform average weights. From table 6, we observe that DB-MPA achieves the largest improvement, while DB-MPA-LS performs slightly worse but reduces inference cost to essentially the same unit-time speed as SD v1.5. In contrast, the baseline RS, despite interpolating the same 2–4 reward-basis models, performs substantially worse than DB, and its performance degrades noticeably as the number of rewards increases.

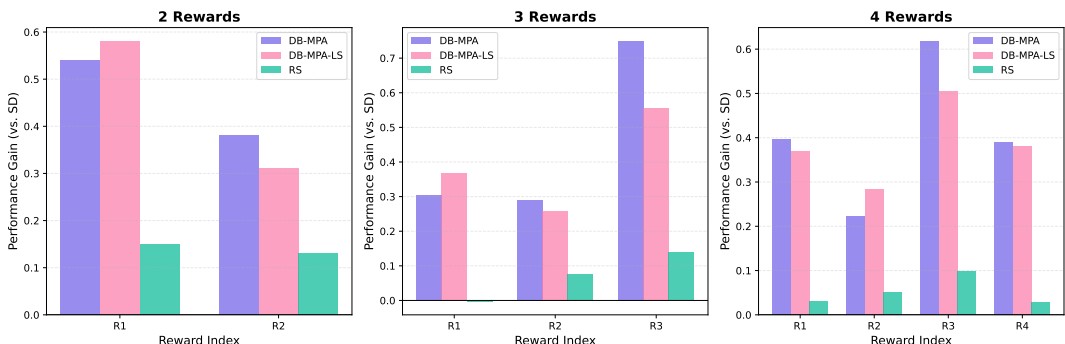

Figure 7: Performance comparison on Short-drawbench 1k test prompts of DB-MPA and baseline algorithms under different numbers of reward models. R1 = ImageReward, R2 = VILA, R3 = Compressibility, R4 = PickScore. Performance improvement is computed as (algorithm reward) - (SD-v1.5 reward). DB-MPA and DV-MPA-LS consistently outperform RS as the number of rewards increases.

Table 6: Performance improvements under 2, 3, and 4 reward settings. Each column shows $\Delta r_i = r_i^{\text{method}} - r_i^{\text{SDv1.5}}$, and the group-wise mean.

|  | 2-Reward | | | 3-Reward | | | | 4-Reward | | | | |
|---|---|---|---|---|---|---|---|---|---|---|---|---|
| **Method** | $\Delta r_1$ | $\Delta r_2$ | Avg | $\Delta r_1$ | $\Delta r_2$ | $\Delta r_3$ | Avg | $\Delta r_1$ | $\Delta r_2$ | $\Delta r_3$ | $\Delta r_4$ | Avg |
| DB | 0.54 | **0.38** | **0.46** | 0.30 | **0.29** | **0.75** | **0.45** | **0.40** | 0.22 | **0.62** | **0.39** | **0.41** |
| LS | **0.58** | 0.31 | 0.45 | **0.37** | 0.26 | 0.56 | 0.39 | 0.37 | **0.28** | 0.50 | 0.38 | 0.38 |
| RS | 0.15 | 0.13 | 0.14 | -0.00 | 0.08 | 0.14 | 0.07 | 0.03 | 0.05 | 0.10 | 0.03 | 0.05 |

## C.5 RESULTS FOR THREE-REWARD SETTING

To evaluate DB-MPA in a more complex multi-objective setting, we conducted experiments using three distinct reward functions: ImageReward for text-image alignment, VILA for aesthetic quality, and PickScore Kirstain et al. (2023) as a proxy for human preference w. We evaluate the result on a smaller test set with 25 prompts generated by GPT-4 (Achiam et al., 2023) with identical 30 fixed random seeds for each prompt. The full list of this smaller test set is in our code. Figure 8 illustrates DB-MPA performance across various weight combinations in the three-reward setting. DB-MPA consistently adapts its outputs to reflect user-specified preferences, demonstrating scalable control without requiring additional retraining.

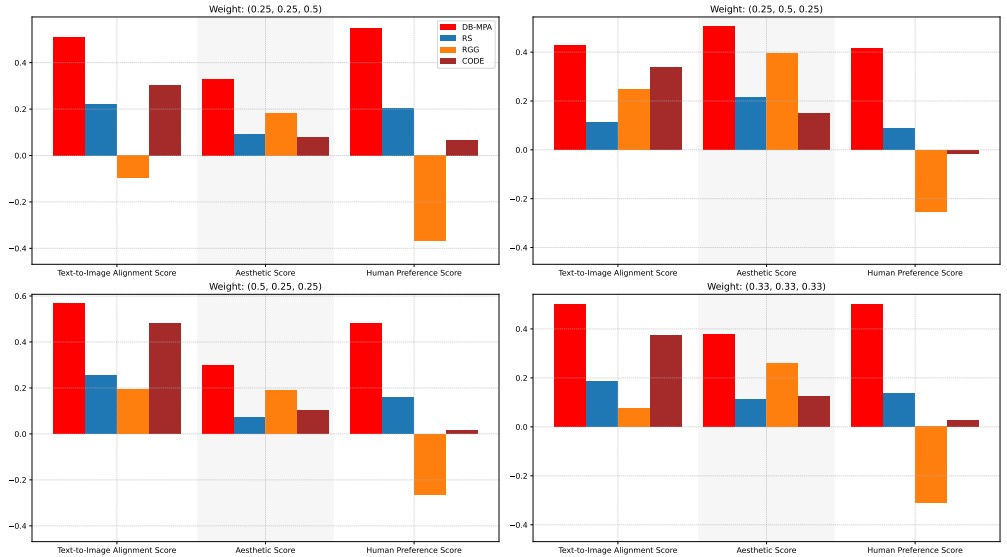

Figure 8: Performance comparison of DB-MPA and baseline algorithms in the $m = 3$ rewards setting, using ImageReward (text-image alignment), VILA (aesthetic quality), and PickScore (human preference). Each bar shows the improvement over SDv1.5 for the corresponding reward. DB-MPA consistently outperforms all baselines across different weight combinations and all reward dimensions.

## C.6 EXTENSION TO SDXL

To examine whether the preference trade-off behavior observed in SD 1.5 carries over to a substantially larger backbone, we extend our experiments to Stable Diffusion XL (SDXL). The SDXL base UNet contains 1.6B parameters, more than 5× larger than the 300M UNet in SD 1.5, and the full SDXL pipeline totals approximately 2.6B parameters. Following seminal RL works on fine-tuning diffusion model Fan et al. (2023); Black et al. (2024) that train and validate on a single prompt, we also fine-tune SDXL on the prompt ("an orange colored sandwich"). As in the SD-1.5 setup, we use two reward models—ImageReward (alignment) and VILA (aesthetics). The corresponding training reward curves are shown in Figure 9.

After training, we evaluated the model across different preference weights. For DB-MPA, DB-MPA-LS, and two baselines RS and CoDe, we swept $w$ in increments of 0.1. For RGG, due to it huge inference time cost, we only evaluated five points: $w \in \{0, 0.2, 0.5, 0.7, 1.0\}$. For each point, 64 random seeds are used. The resulting Pareto front is shown in Figure 10. Table 7 reports the numerical results for the representative weights. The qualitative trends are consistent with our observations in SD 1.5: DB-MPA and DB-MPA-LS continue to provide controllable reward trade-offs in the SDXL setting, achieving larger performance improvements compared to other baselines. Training-based methods (DB-MPA, DB-MPA-LS, and RS) demonstrate superior performance over training-free approaches (RGG and CoDe). Notably, CoDe exhibits similar performance across different reward weightings, which may be attributed to SDXL's larger VAE being more sensitive to noise in the intermediate-step predicted images during CoDe's lookahead best-of-N sampling scheme.

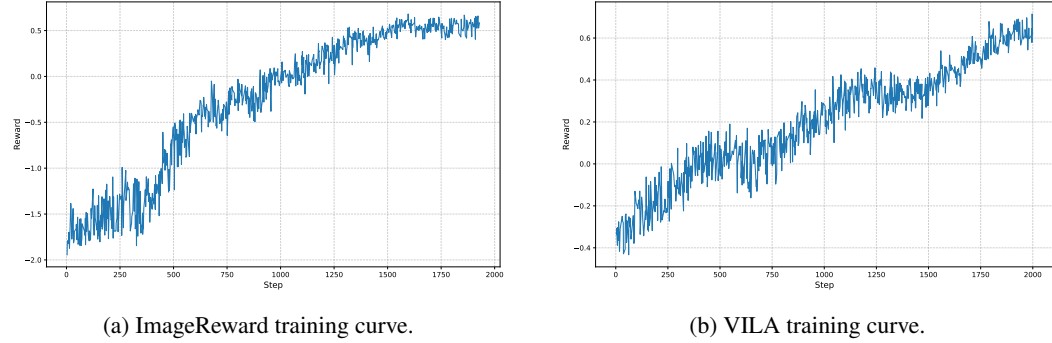

(a) ImageReward training curve.       (b) VILA training curve.

Figure 9: SDXL training curves for the two reward models. Training was run for roughly 2000 epochs over 72 GPU(A100) hours.

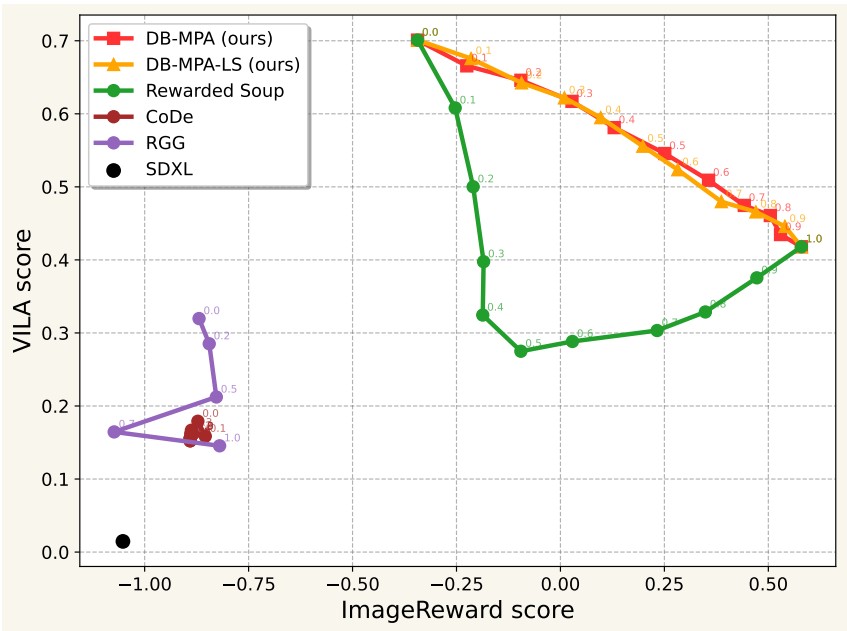

Figure 10: Pareto front of the SDXL LoRA model across preference weights.

Table 7: SDXL single prompt results: $r_1$ = ImageReward, $r_2$ = VILA. The pretrained SDXL has: $r_1 = -1.05$, $r_2 = 0.01$.

|  |  | DB-MPA | | DB-MPA-LS | | RS | | CoDe | | RGG | |
|---|---|---|---|---|---|---|---|---|---|---|---|
|  |  | $r_1$ | $r_2$ | $r_1$ | $r_2$ | $r_1$ | $r_2$ | $r_1$ | $r_2$ | $r_1$ | $r_2$ |
| Reward (↑) | $w=0.0$ | $-0.34$ | 0.70 | $-0.34$ | 0.70 | $-0.34$ | 0.70 | $-0.87$ | 0.18 | $-0.87$ | 0.32 |
|  | $w=0.2$ | $-0.10$ | 0.65 | $-0.09$ | 0.64 | $-0.21$ | 0.50 | $-0.89$ | 0.16 | $-0.84$ | 0.29 |
|  | $w=0.5$ | 0.25 | 0.55 | 0.20 | 0.56 | $-0.10$ | 0.27 | $-0.89$ | 0.16 | $-0.83$ | 0.21 |
|  | $w=0.7$ | 0.44 | 0.47 | 0.39 | 0.48 | 0.23 | 0.30 | $-0.89$ | 0.15 | $-1.07$ | 0.16 |
|  | $w=1.0$ | 0.58 | 0.42 | 0.58 | 0.42 | 0.58 | 0.42 | $-0.88$ | 0.16 | $-0.82$ | 0.15 |

## C.7 VISUAL SIMILARITY OF DB-MPA AND DB-MPA-LS

We present visual comparisons among DB-MPA, DB-MPA-LS, and RS under the two-reward setting using ImageReward (text-image alignment) and VILA (aesthetic quality). The results indicate that, for interpolating the same pair of diffusion reverse processes, DB-MPA and DB-MPA-LS yield visually similar outputs, both surpassing the baseline RS in image quality.

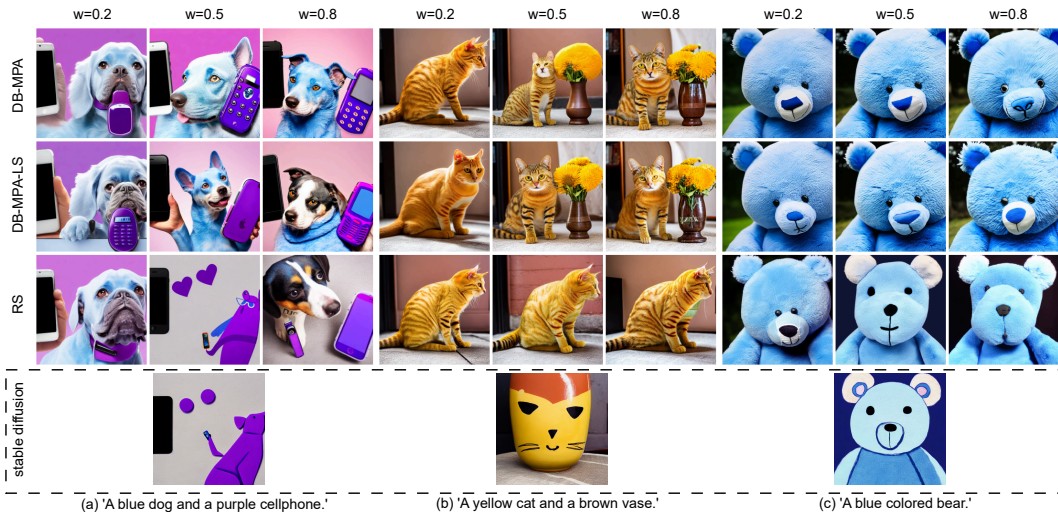

Figure 11: $w$ denotes the weight assigned to ImageReward, with $1-w$ corresponding to the weight for VILA. Both DB-MPA and DB-MPA-LS produce visually similar results, and each generates images better aligned to the user's preference than the baseline RS.

## C.8 VISUAL COMPARISON WITH BASELINES

We provide additional qualitative comparisons between DB-MPA and the baselines in fig. 12 using prompts from both the train and test sets, for $w \in \{0.2, 0.5, 0.8\}$. Despite requiring no extra fine-tuning, DB-MPA generates images that are visually close to the MORL oracle baseline, and outperforms all other baseline methods.

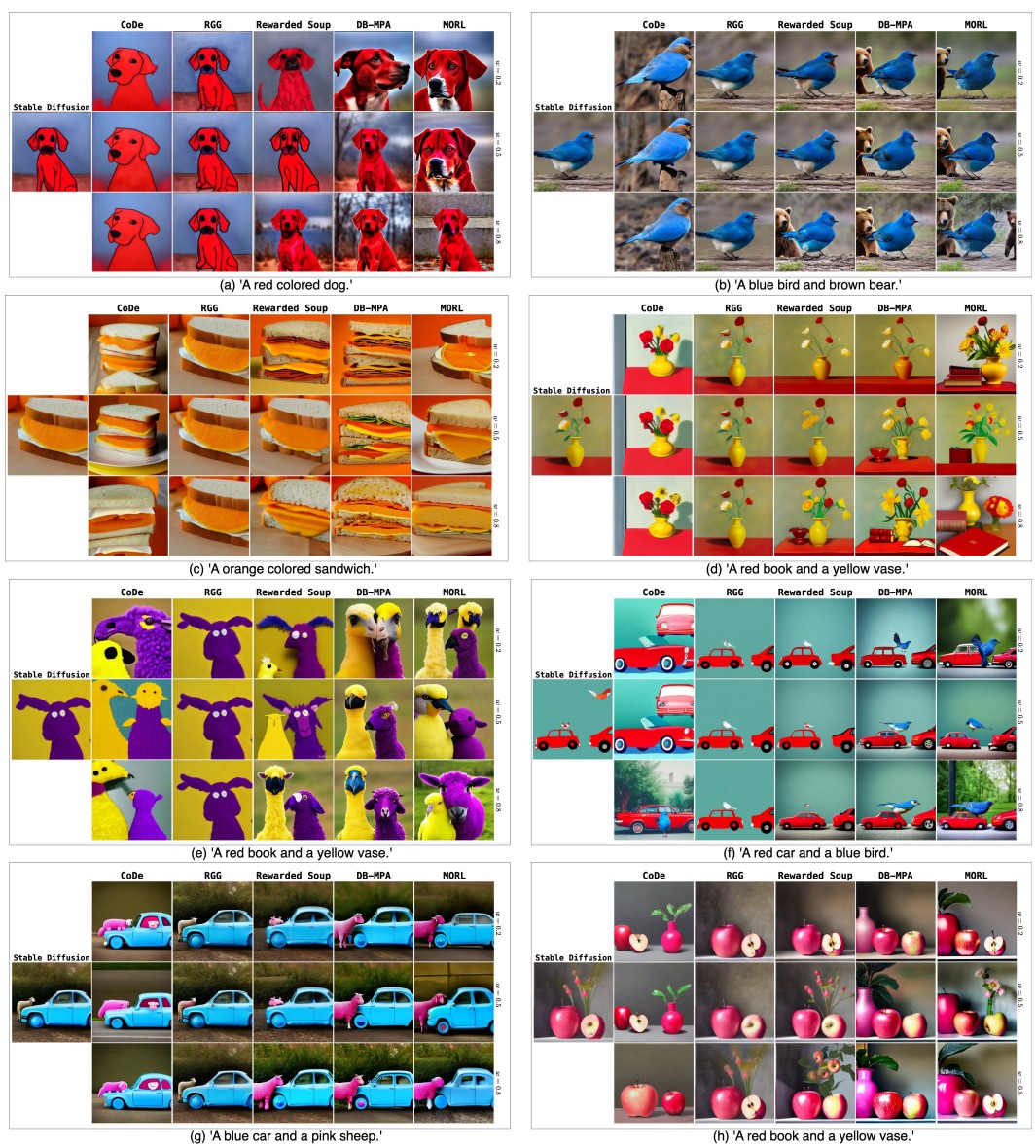

Figure 12: Qualitative comparison of DB-MPA with baselines. Subfigures (a)–(d) correspond to training prompts, and (e)–(h) to test prompts. In several cases, such as (a) and (e), Stable Diffusion produces cartoonish or unrealistic outputs. In contrast, DB-MPA generates more realistic and semantically aligned images by effectively leveraging multi-reward alignment, without requiring any additional fine-tuning.

## C.9  Multi-Preference Alignment with Finer Granularity

In Fig. 13, we present additional results of DB-MPA and RS for multi-preference alignment with finer granularity of $w$, with $w \in \{0.1, 0.2, \ldots, 0.9\}$. As observed, both algorithms exhibit a smooth transition from aesthetically pleasing results to outputs that are more aligned with the input prompt. However, DB-MPA typically achieves better alignment with the input prompt, especially for $w \in [0.3, 0.7]$.

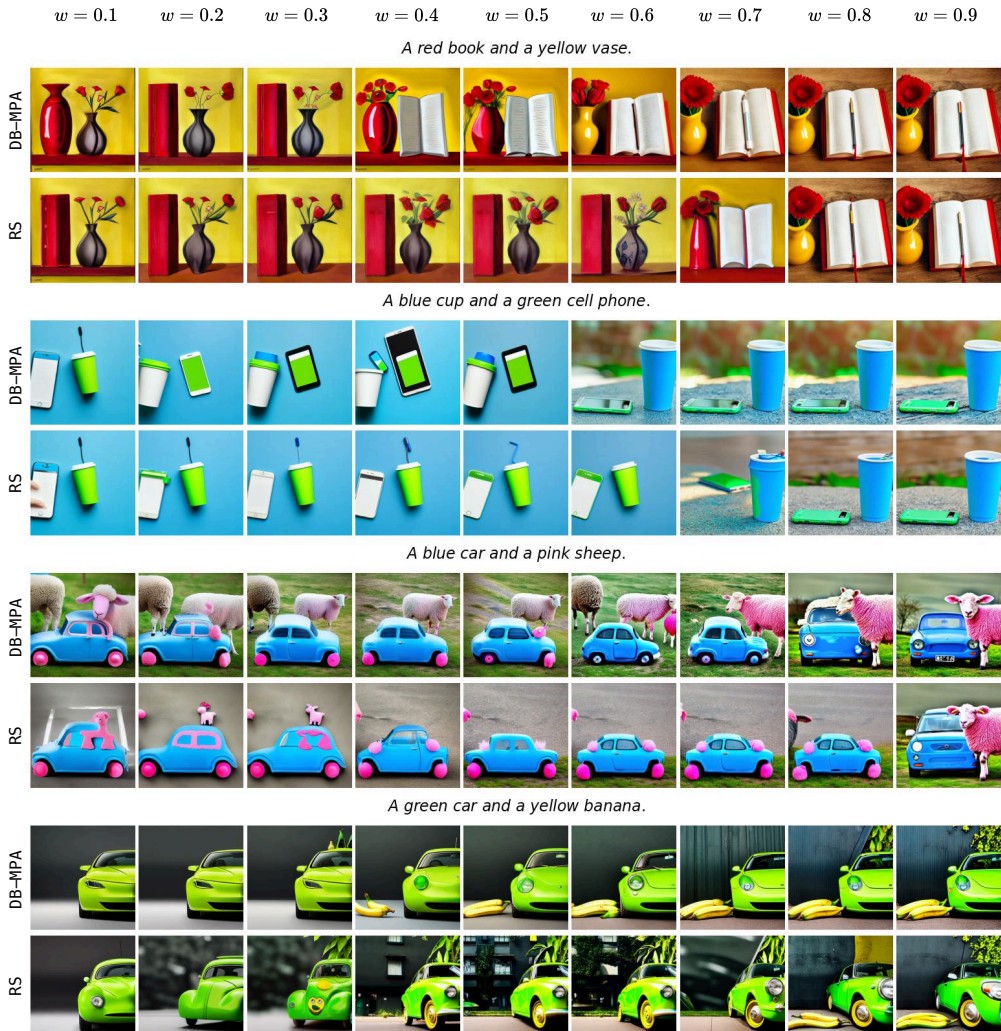

Figure 13: Qualitative comparison between DB-MPA (first row) and RS (second row) with ImageReward and aesthetic score as rewards. Each block shows generations as the ImageReward weight $w$ increases from 0.1 to 0.9 (left to right). The first two examples are from the training prompt set, and the last two from the test prompt set. DB-MPA demonstrates smoother transitions and more precise alignment with the target reward preferences compared to RS, supporting the trends observed in the quantitative results.

# D DB-KLA: ADDITIONAL RESULTS

This section presents additional results for DB-KLA, including qualitative comparisons with the MORL oracle across diverse prompts. It also evaluates DB-KLA's controllability under fine-grained variations of the KL weight.

## D.1 QUALITATIVE COMPARISON WITH BASELINES

We provide additional qualitative comparisons between DB-KLA and the MORL oracle baseline in fig. 14, using prompts from both the train and test sets. These examples show how DB-KLA adapts generation quality as the KL regularization strength changes, producing outputs that closely resemble those of the oracle baseline.

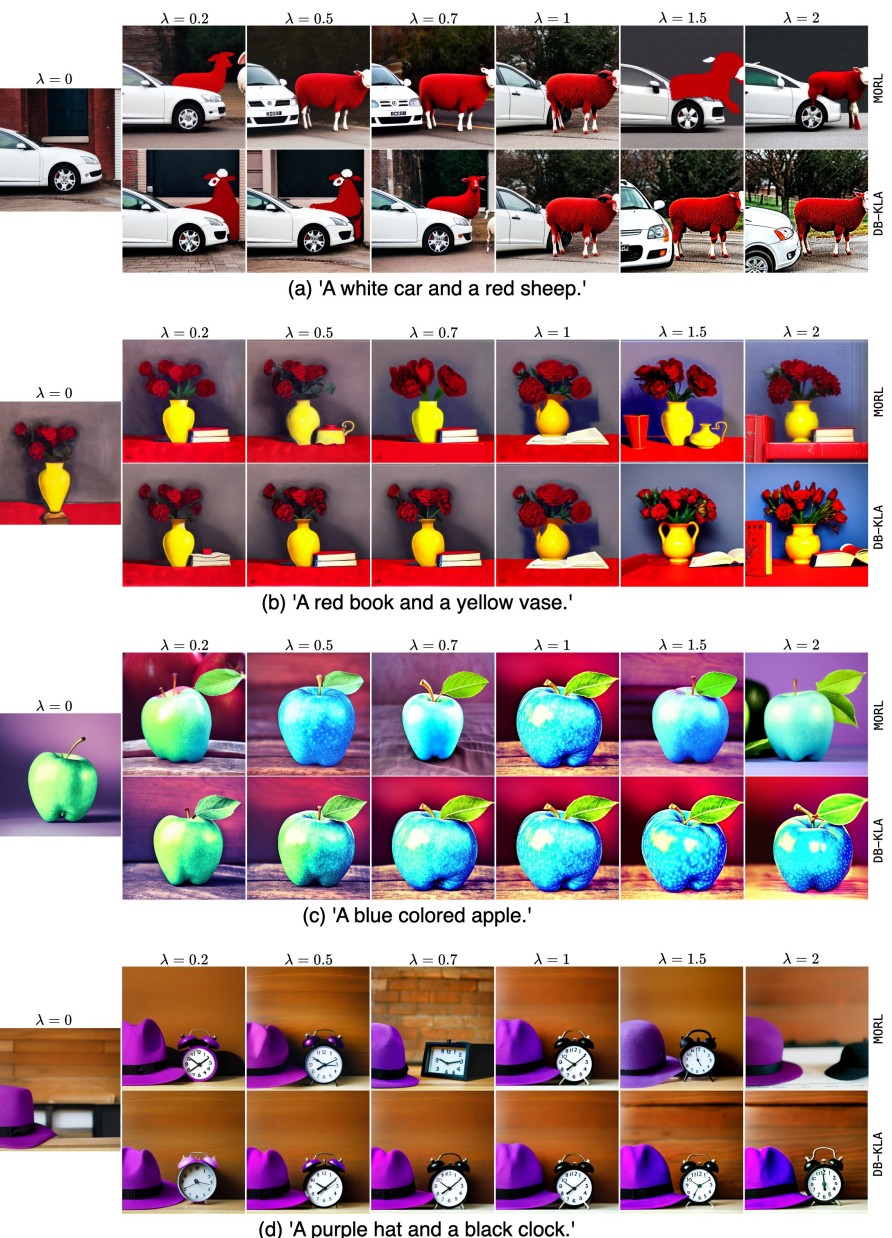

(a) 'A white car and a red sheep.'

(b) 'A red book and a yellow vase.'

(c) 'A blue colored apple.'

(d) 'A purple hat and a black clock.'

Figure 14: Qualitative comparison between DB-KLA and MORL for KL weights $\lambda \in \{0.2, 0.5, 0.7, 1.0, 1.5, 2.0\}$. The first two rows show the results with train prompts, and the last two show the results with test prompts. DB-KLA generates images of similar quality to those of the MORL oracle baseline without any additional fine-tuning.

## D.2 KL ALIGNMENT WITH FINER GRANULARITY

In Fig. 15, we present additional results of DB-KLA with finer granularity of $\lambda$, for $\lambda = [0, 0.2, 0.5, 0.7, 1.0, 1.5, 2.0]$. As regularization increases, the model shifts more toward optimizing the text-to-image reward, producing images that better match the prompt but drift further from the original Stable Diffusion output.

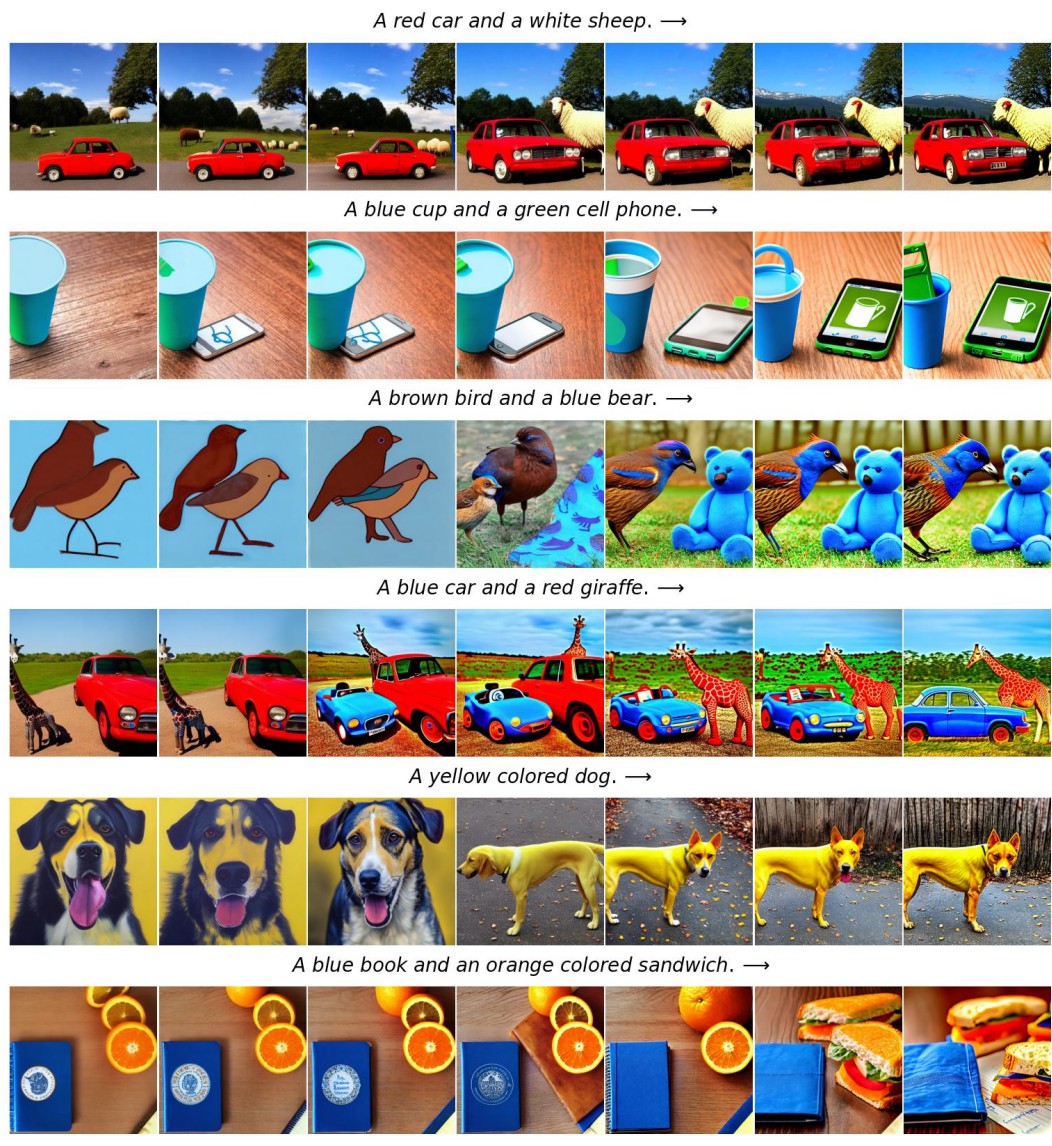

Figure 15: KL weight alignment in DB for $\lambda = [0, 0.2, 0.5, 0.7, 1.0, 1.5, 2.0]$, with $\lambda$ increasing from left to right.

## E IMPACT STATEMENT

Diffusion Blend enables flexible, inference-time alignment of diffusion models with user-specified preferences over multiple reward objectives and regularization strengths, without additional fine-tuning. This significantly reduces computational costs and increases adaptability for personalization. By leveraging a small set of fine-tuned models, it opens the door to scalable, user-controllable generative AI and sets the stage for more preference-aware deployment. While our work has broad implications for AI alignment and deployment as it enhances the existing diffusion models' performance, we do not foresee any immediate societal concerns that require specific highlighting.

# F    THE USE OF LARGE LANGUAGE MODELS

Portions of this work were prepared with the assistance of a large language model (ChatGPT, GPT-5, by OpenAI). The model was used as a writing assistant to improve clarity, grammar, and organization of the manuscript, and to suggest alternative phrasings of technical content written by the authors. All ideas, experiments, analyses, and final decisions regarding the content remain the responsibility of the authors. The test prompt set for the Drawbench prompt dataset is generated by LLM, following the standard benchmark like GenEval. The model was not used to generate research ideas, perform experiments, or create unverifiable scientific claims.

