# OpenReview forum: "Diffusion Blend: Inference-Time Multi-Preference Alignment for Diffusion Models"
_ICLR.cc/2026/Conference — ICLR 2026 Poster_

### Official Review · Reviewer_PHNS · 2025-10-30

**Soundness:** 4
**Presentation:** 3
**Contribution:** 4
**Rating:** 8
**Confidence:** 4

**Summary:**

This is a classifier‑free guidance–like formulation designed for reward alignment in a score-based generated model.

**Strengths:**

Overall, the paper mainly makes a theoretical contribution.

## Presentation: ~75th percentile

This paper is comprehensible to diffusion model researchers, but it possesses some weaknesses in presentation.

## Soundness: 80~90th percentile

A great portion of the soundness stems from the reliability and the success of CFG. The formulation appears correct based on my understanding of CFG and score-based SDE.

## Contribution: ~90th percentile

To the best of my knowledge, you are the first, aside from any concurrent work, to propose this
formulation. Your contribution is particularly strong because it is presented within a score‑based SDE
framework. It could have been even more impactful had you expressed it in Karras’s SDE, although that is
not strictly necessary.

## Note
I hope the AC is aware that the rating is calibrated using percentiles to reduce evaluation noise effectively.

**Weaknesses:**

## Presentation

Presenting advanced mathematics elegantly is always a challenge, and a successful example in writing I’ve read is the score-based SDE by Yang et al. Perhaps it would be better to present the main idea without too many interruptions from details, making the mathematical notation less daunting in the main paper for a wider range of readers.  But that is impossible without a significant modification of the manuscript.  If accepted, I recommend rewriting your final manual script or, if any, your preprints.

There are some minor formatting issues, such as inconsistent use of italics and bold across sections, and the abuse of italics applied to the entire paragraph.

**Questions:**

As noted in strength, I prefer to see the reformulation written in Karra's SDE instead.

---

> ### Author Response · Authors · 2025-11-24
> **Response to Reviewer PHNS**
>
> We thank the reviewer for their encouraging comments on the presentation, soundness, and contribution of the paper. Based on the reviewer's suggestion, **we have now added new results: (i) scaling the experiments to a larger diffusion model (SDXL), and (ii) adding a number-of-reward scaling study (until $m=4$ rewards) to evaluate performance gain under increasing objectives**. The results and discussions are given in **Appendix C.6 and C.4** of the revised manuscript. Please note that all changes in the revised manuscript are highlighted in blue. We hope that these additions will further clarify the novelty and effectiveness of our contributions. Below, we address all the questions and comments raised by the reviewer.
>
> **Q1. Improvement on presentation is needed.**
>
> **Response:**
> We thank the reviewer for the constructive comments on presentation and the pointer to Yang et al. (score-based SDE tutorial) as an example. We will revise the camera-ready version to improve clarity, restructuring the theoretical sections for better readability, and fixing the formatting inconsistencies (e.g., italics/bold usage).
>
> **Q2. Reformulation written in Karra’s SDE**
>
> **Response:**
> We appreciate the reviewer’s suggestion. The EDM formulation (Karras’s SDE) introduces a general SDE that decouples the scheduler from the training loss and rewrites both the forward and reverse processes. Our derivation is fully compatible with this EDM/Karras formulation. The core idea of our framework starts from the general reverse SDE of a pretrained model (Eq. (2)): after optimizing the KL-regularized reward-maximization objective (Eq. (3)), the underlying marginal distribution is modified, but the form of the reverse SDE remains the same as Eq. (2), except for the altered score term (Proposition 1). This modified score admits an empirically efficient approximation (Eq. (8)) that is linear in the KL refactor $\lambda$ and the reward weights given the reward-basis models, from which our algorithm is derived.
>
> These conclusions continue to hold under the Karras SDE as:
>
> 1. Proposition 1 also holds for the Karras SDE. We provide a general version (Appendix A.1 Proposition 3) of Proposition 1, along with additional discussion in Remark 3 (line 756-766) regarding the general noise-injecting process.
>
> 2. The general EDM SDE takes the form $dx_t = (f_1(t)+f_2(t)\nabla_xp_t(x))dx+f_3(t)d\omega_t$ (Eq. (6) of [1]), where the drift depends linearly on the score. Since our derivation of Proposition 2 relies solely on this linear score structure, Proposition 2 applies directly to the EDM/Karras formulation without any modification.
>
> [1] Karras, Tero, et al. "Elucidating the design space of diffusion-based generative models." Advances in neural information processing systems 35 (2022)

---

> > ### Comment · Reviewer_PHNS · 2025-11-25
> >
> > I have no other questions. Good work.

---

> > > ### Author Response · Authors · 2025-11-26
> > >
> > > Thanks! We are glad that your concern got resolved.

---

### Official Review · Reviewer_bBuv · 2025-10-30

**Soundness:** 4
**Presentation:** 4
**Contribution:** 3
**Rating:** 8
**Confidence:** 4

**Summary:**

This paper proposes Diffusion Blend that aligns diffusion models' generation process with user-specified multi-preferences. Crucially, this is done at inference time such that users can dynamically control preferences as they wish. Diffusion Blend trains different versions of diffusion models that maximize the given reward functions and then combine their scores during the generation process. This paper introduces theoretical justification for such a process and demonstrates its effectiveness in quantitative and qualitative experiments.

**Strengths:**

- Importance of the problem: Dynamically aligning diffusion models' generation process with user preferences is a challenging problem that can have a great impact on content creation applications by providing users with knobs to adjust their content.
- This paper proposes a simple algorithm that achieves this purpose by combining scores of diffusion models trained for specific rewards. This is theoretically justified well, and the simplicity of the algorithm would enable easy adoption of the method.
- Maintaining multiple copies of diffusion models could be heavy in storage and computation, but the experiment with LoRA shows it is possible to achieve efficient alignment by maintaining relatively small copies.
- The evaluation in the experiments section shows a consistent boost in performance over the baselines considered in the paper.

**Weaknesses:**

- The experiments are done with a relatively small number of reward models (most with two) and a single class of diffusion model, Stable Diffusion. The results could have been stronger by pushing the limit with more rewards (e.g. 6~10) and with more recent diffusion models such as Flux.
- In practice, LoRA-based diffusion model weight combination techniques (e.g. Zou, Shen, Bouganis, and Zhao, ICLR 2025) could be a strong candidate to achieve the same purpose. How does it perform, and is there a practical advantage to using Diffusion Blend over it?

Minor: in equation 11) $\log p^{tar}_t$ in the right hand side needs to be  $\log p^{pre}_t$?

**Questions:**

It would be great if the rebuttal could comment on the points in the weaknesses section.

---

> ### Author Response · Authors · 2025-11-24
> **Response to Reviewer bBuv**
>
> We thank the reviewer for their encouraging comments regarding the *“importance of the problem”*, that our approach is *“theoretically justified well, and the simplicity of the algorithm would enable easy adoption of the method”*, and that the *“evaluation in the experiments section shows a consistent boost in performance over the baselines”*.
>
> Based on the reviewer's suggestion, **we have now added new results: (i) scaling the experiments to a larger diffusion model (SDXL), and (ii) adding a number-of-reward scaling study (until $m=4$ rewards) to evaluate performance gain under increasing objectives**. The results and discussions are given in **Appendix C.6 and C.4** of the revised manuscript. Please note that all changes in the revised manuscript are highlighted in blue. We hope that these additions will further clarify the novelty and effectiveness of our contributions. Below, we address all the questions and comments raised by the reviewer.
>
> **Q1. The results could have been stronger … with more recent diffusion models.**
>
> **Response:**
> According to the reviewer’s suggestion, we have now included new experimental results using SDXL as the base model; please see Appendix C.6. These results confirm that our approach transfers effectively to larger diffusion backbones and outperforms all baseline approaches.
>
> **Q2. The results could have been stronger by pushing the limit with more rewards …**
>
> **Response:**
> Our original submission already incorporated four reward models with the details in Appendix C.3 and a 3-reward experiment in Appendix C.5. Based on the reviewer’s suggestion, we have now added new experimental results that more clearly show the effect of increasing the number of rewards; please see Appendix C.4. These results again confirm that our methods, including the inference-time efficient DB-MPA-LS variant, continue to outperform as $m$ grows.
>
> **Q3. Comparison with LoRA-based diffusion model weight combination techniques (e.g., Zou, Shen, Bouganis, and Zhao, ICLR 2025)?**
>
> **Response:**
> As discussed in  Related Work (Section 2), multi-LoRA fusion methods focus on concept composition using heuristic scheduling. Our method is fundamentally different in three ways:
>
> 1. Problem setting: Multi-LoRA (Zhong et al., 2024a) addresses multi-concept generation by linearly interpolating LoRA weights trained on single fixed concepts with static, uniform coefficients. DB instead tackles inference-time multi-preference alignment: the user specifies an arbitrary weight vector $w$ at run time, and our DB-MPA adjusts the blend to satisfy those arbitrary preferences.
>
> 2. Algorithmic machinery: LoRA-Composite and LoRA-Switch from (Zhong et al., 2024a) are heuristic weight merges, where LoRA-Composite works with fixed uniform weights for each SFT single-concept LoRA and LoRA-Switch alternates between different LoRAs. Both methods enforce positive weight constraints and cannot accommodate the negative weights required by our DB-KLA variant, while DB introduces two new algorithms, DB-MPA and DB-KLA, that (i) take arbitrary user-specified weights as input and (ii) extend naturally to negative weights through a principled KL-adjusted weight.
>
> 3. In-depth theoretical analysis: We derive our blend rule from the SDE formulation of diffusion sampling, proving that it targets the distribution. Such analysis is absent in existing multi-LoRA work, which provides no guarantee beyond empirical success.

---

> > ### Comment · Reviewer_bBuv · 2025-11-27
> >
> > I thank the authors for the rebuttal, and I maintain this is a good work and thus my rating.

---

### Official Review · Reviewer_b58F · 2025-10-31

**Soundness:** 3
**Presentation:** 4
**Contribution:** 4
**Rating:** 6
**Confidence:** 4

**Summary:**

This paper tackles the challenge of aligning diffusion models with multiple, potentially conflicting objectives at inference time. Existing RL-based alignment methods typically optimize a single reward under fixed KL regularization, which limits flexibility. The authors introduce Diffusion Blend, a novel framework that enables inference-time multi-preference alignment, which allow the model to generate outputs based on any user-specified combination of reward functions and regularization strengths without additional fine-tuning. They instantiate this idea with three algorithms: DB-MPA (for multi-reward alignment), DB-KLA (for controllable KL regularization), and DB-MPA-LS (a low-cost approximation). Empirical results demonstrate that these methods outperform existing baselines and perform comparably to models fine-tuned individually, offering a practical way to achieve user-driven, flexible alignment during inference.

**Strengths:**

1. The paper is well written and easy to follow.

2. The studies of conducting inference time alignment on diffusion models are novel and interesting.

3. The authors conduct extensive experiments to verify the effectiveness of thier method.

**Weaknesses:**

1. It's better for authors to have some results on larger models like SDXL to further prove the effectiveness of thier method.

2. It's better to demonstrate that the model can be used in wide applications like image editing.

3. The authors use DPOK for fine-tuning models, is this method sensitive to different RL algorithms?

**Questions:**

Please refer to the answer in Weaknesses part.

---

> ### Author Response · Authors · 2025-11-24
> **Response to Reviewer b58F**
>
> We thank the reviewer for their encouraging comments that *“paper is well written and easy to follow”*, *“inference time alignment on diffusion models are novel and interesting”*, and *“authors conduct extensive experiments to verify the effectiveness”*.
>
> Based on the reviewer's suggestion, **we have now added new results: (i) scaling the experiments to a larger diffusion model (SDXL), and (ii) adding a reward-scaling study (2 → 3 → 4 rewards) to evaluate performance gain under increasing objectives**. The results and discussions are given in **Appendix C.6 and C.4** of the revised manuscript. Please note that all changes in the revised manuscript are highlighted in blue. We hope that these additions will further clarify the novelty and effectiveness of our contributions. Below, we address all the questions and comments raised by the reviewer.
>
> **Q1. More results on larger models like SDXL**
>
> **Response:**
> According to the reviewer’s suggestion, we have now included **new experimental results using SDXL as the base model**; please see Appendix C.6. These results confirm that our approach transfers effectively to larger diffusion backbones and outperforms all baseline approaches.
>
> **Q2. It's better to demonstrate that the model can be used in wide applications like image editing.**
>
> **Response:**
> We appreciate the suggestion. However, image editing requires task-specific architectures and evaluation protocols, which are beyond the scope of our work. Our focus is inference-time preference alignment, and we leave extensions such as image editing for future work.
>
> **Q3. The authors use DPOK for fine-tuning models; is this method sensitive to different RL algorithms?**
>
> **Response:**
> Our method is agnostic to the specific RL fine-tuning algorithm. Diffusion Blend only requires access to KL-regularized reward-aligned models and can therefore be combined with any RL method that optimizes a KL-regularized objective. We use DPOK in our experiments because it is currently the most established and widely adopted KL-regularized fine-tuning algorithm for diffusion models (Fan et al., 2023), but our framework is not tied to it.

---

> > ### Comment · Reviewer_b58F · 2025-11-27
> >
> > Thanks for your reply! I think the authors have resolved my concerns and I will maintain my positive score for this paper.

---

### Official Review · Reviewer_RqHN · 2025-11-03

**Soundness:** 3
**Presentation:** 2
**Contribution:** 2
**Rating:** 4
**Confidence:** 4

**Summary:**

This paper addresses the problem of inference-time multi-preference alignment for diffusion models, where the users can adjust reward trade-offs and regularization strengths on-the-fly without additional finetuning. The DiffusionBlend pipeline blends the backward diffusion trajectories of multiple RL-finetuned models to distinct reward functions. The method is theoretically motivated and empirically validated on a series of benchmarks and alignment tasks.

**Strengths:**

1. The problem is novel and well-motivated. The paper formalizes the inference-time multi-preference alignment problem from the perspective of MORL, which is practically important and underexplored in diffusion literature.

2. The approach is theoretically justified with clear derivations and bounds on approximation errors.

3. Strong empirical results on multiple datasets and reward functions.

**Weaknesses:**

1. The Jensen-gap approximations in Eq. 8 are only empirically validated via downstream metrics; direct error analysis on $ \Delta(r,\alpha) $ (beyond Appendix bounds) would strengthen claims.

2. The paper builds on KL-regularized RL fine-tuning for diffusion models, but does not sufficiently discuss or compare to DiffusionDPO (Wallace et al., 2024), DDPO (Black et al., 2024), etc, which are dominant lines of work in this space. For example, the author cites DPO (Rafailovetal., 2023) for diffusion model alignment in section 4, but does not mention DiffusionDPO (Wallace et al., 2024), which is the direct diffusion analog and is relevant.

2. The training cost still scales linearly with respect to the number of rewards. This can be costly in applications.

3. Based on the provided pages, evaluations focus on two rewards (alignment + aesthetics) and Stable Diffusion v1.5. Scaling to m > 2 or diverse rewards (e.g., human preferences via PickScore, diversity) and larger base models (e.g., SDXL) is not shown.

4. Strong compared to RS/CoDe/RGG, but no comparison to DPO-inspired diffusion variants (e.g., Calibrated DPO) or recent multi-LoRA fusion methods (Zhong et al., 2024a). The best individual RL-finetuned model per w (the oracle) is also missing.

**Questions:**

see weaknesses.

---

> ### Author Response · Authors · 2025-11-24
> **Response to Reviewer RqHN (1/2)**
>
> We thank the reviewer for their encouraging comments that *"problem is novel and well-motivated... practically important and underexplored in diffusion literature"*, *"approach is theoretically justified with clear derivations and bounds on approximation errors"*, and the paper has *"strong empirical results on multiple datasets and reward functions"*.
>
> Based on the reviewer's suggestion, **we have now added new results: (i) scaling the experiments to a larger diffusion model (SDXL), and (ii) scaling the experiments to a four-reward functions setting**. The results and discussions are given in **Appendix C.6 and C.4** of the revised manuscript. Please note that all changes in the revised manuscript are highlighted in blue. We hope that these additions will further clarify the novelty and effectiveness of our contributions. Below, we address all the questions and comments raised by the reviewer.
>
> **Q1. Scaling larger base models (e.g., SDXL) is not shown**
>
> **Response:**
> According to the reviewer’s suggestion, we have now included new experimental results using SDXL as the base model; please see Appendix C.6. These results confirm that our approach transfers effectively to larger diffusion backbones and outperforms all the baseline approaches.
>
> **Q2. Scaling to m > 2 or diverse rewards is not shown.**
>
> **Response:**
> Please note that our original submission already included experiments with m = 4 reward models; see Section 5 (lines 348–355) with the details in Appendix C.3 and 3-reward experiment Appendix C.5. Based on the reviewer’s suggestion, we have now added new experimental results that more clearly show how performance scales with the number of rewards; please see Appendix C.4. These results again confirm that our methods—including the inference-time efficient DB-MPA-LS—continue to outperform baselines as m increases.
>
> **Q3. The Jensen-gap approximations in Eq. 8 are only empirically validated via downstream metrics; direct error analysis on Δ(r, a) (beyond Appendix bounds) would strengthen claims.**
>
> **Response:**
> We would like to clarify that we already provide a detailed theoretical analysis of the Jensen-gap approximation error in Appendix A.2. In particular, we establish an explicit upper bound $ \lVert \Delta_t(r,a) \rVert_2 \leq L_{t,1}\times L_{t,2}+L_{t,3}$, and explain the conditions under which each term $L_{t,\cdot}$ can become negligible. This bound is relatively tight: under a mild assumption on the reward function proposed in prior work (see line 870), the entire upper bound collapses to zero, meaning the approximation becomes exact in that setting.
>
> **Q4. Comparison with DiffusionDPO and DDPO**
>
> **Response:**
> We want to first clarify that our contribution is a general inference-time framework that blends single-reward with KL regularization optimized diffusion models and jointly handles both rewards reweighting and KL reweighting, which is compatible with any KL-regularized RL finetuning method, including DDPO and DPO. **Our approach (DB) is therefore orthogonal to RL fine-tuning methods used.** We use DPOK because it is the most established approach that explicitly optimizes a KL-regularized objective in diffusion models, allowing clearer comparison across baselines, especially the RL retraining baseline (MORL) in both sets of our experiments: multi preference alignment, and KL reweighting alignment. DDPO does not incorporate KL regularization, which is essential in our setting, and DiffusionDPO/DPO require pairwise preference data to avoid reward modeling, whereas explicit reward functions are available in our problem setting. Therefore, we chose DPOK to represent the single reward maximization RL algorithm. But we believe blended models finetuned with DDPO or DPO can also benefit from the same inference-time improvements of DB.
>
> **Q5. The training cost scales linearly with respect to the number of rewards.**
>
> **Response:**
> In multi-objective alignment, the theoretical optimal approach is RL retraining at every preference point $w$, which incurs an unbounded (infinite in the continuous limit) training cost. In contrast, our linear $\times m$ cost (one RL run per basis reward) is orders of magnitude more efficient. Moreover, as shown in Table 1, the training-free baselines (CoDe and RGG) impose a very large inference overhead, making them impractical for real-time or interactive use.

---

> > ### Author Response · Authors · 2025-11-24
> > **Response to Reviewer RqHN (2/2)**
> >
> > **Q6. Comparison with multi-LoRA fusion methods (Zhong et al., 2024a)**
> >
> > **Response:**
> > As discussed in  Related Work (Section 2), multi-LoRA fusion methods focus on concept composition using heuristic scheduling. Our method is fundamentally different in three ways:
> >
> > 1. Problem setting: Multi-LoRA (Zhong et al., 2024a) addresses multi-concept generation by linearly interpolating LoRA weights trained on single fixed concepts with static, uniform coefficients. DB instead tackles inference-time multi-preference alignment: the user specifies an arbitrary weight vector $w$ at run time, and our DB-MPA adjusts the blend to satisfy those arbitrary preferences.
> >
> > 2. Algorithmic machinery: LoRA-Composite and LoRA-Switch from (Zhong et al., 2024a) are heuristic weight merges, where LoRA-Composite works with fixed uniform weights for each SFT single-concept LoRA and LoRA-Switch alternates between different LoRAs. Both methods enforce positive weight constraints and cannot accommodate the negative weights required by our DB-KLA variant, while DB introduces two new algorithms, DB-MPA and DB-KLA, that (i) take arbitrary user-specified weights as input and (ii) extend naturally to negative weights through a principled KL-adjusted weight.
> >
> > 3. In-depth theoretical analysis: We derive our blend rule from the SDE formulation of diffusion sampling, proving that it targets the distribution. Such analysis is absent in existing multi-LoRA work, which provides no guarantee beyond empirical success.
> >
> >
> > **Q7. The best individual RL-finetuned model per w (the oracle) is also missing.**
> >
> > **Response:**
> > We already include these oracle results (best individual RL-finetuned model), denoted as **MORL**, in **Figure 2b** and **Table 1**. As noted, MORL serves as the upper bound achievable by retraining a dedicated RL model for each $w$. Our method consistently approaches this oracle frontier.

---

### Meta-Review · Area_Chair_xHVY · 2026-01-06

**Summary:**

The paper proposes DiffusionBlend, a framework for aligning diffusion models with multiple, potentially conflicting user preferences at inference time. The method blends the backward diffusion trajectories of multiple models (often fine-tuned via RL/LoRA on specific rewards) to generate outputs that satisfy a user-specified combination of rewards, and regularization strengths without requiring additional fine-tuning. The algorithm is praised for being simple to adopt and empirically effective against baselines like RS, CoDe, and RGG.

**Reviewer Concerns:**

The reviews are generally positive, the theoretical grounding (linking to score-based SDEs and CFG), the novelty of the inference-time control issue, and the simplicity of the methods. However, there are consistent requests for stronger baselines (specifically DiffusionDPO and LoRA merging) and evaluation on larger models (SDXL, Flux) to prove scalability.

**Reviewer Scores:**

Reviewers tends to acceptance, The AC agree with the reviewers. However, the authros should considered the issues about the stronger baselines and evaluation on larger models to prove scalability, this shoud be refined in the final versions.

---

### Decision · Program_Chairs · 2026-01-26

Accept (Poster)